# Identification of rare sequence variation underlying heritable pulmonary arterial hypertension

Stefan Gräf et al.[#]

Pulmonary arterial hypertension (PAH) is a rare disorder with a poor prognosis. Deleterious variation within components of the transforming growth factor-β pathway, particularly the bone morphogenetic protein type 2 receptor (*BMPR2*), underlies most heritable forms of PAH. To identify the missing heritability we perform whole-genome sequencing in 1038 PAH index cases and 6385 PAH-negative control subjects. Case-control analyses reveal significant overrepresentation of rare variants in *ATP13A3, AQP1* and *SOX17*, and provide independent validation of a critical role for *GDF2* in PAH. We demonstrate familial segregation of mutations in *SOX17* and *AQP1* with PAH. Mutations in *GDF2*, encoding a BMPR2 ligand, lead to reduced secretion from transfected cells. In addition, we identify pathogenic mutations in the majority of previously reported PAH genes, and provide evidence for further putative genes. Taken together these findings contribute new insights into the molecular basis of PAH and indicate unexplored pathways for therapeutic intervention.

#A full list of authors and their affiliations appears at the end of the paper.

diopathic and heritable pulmonary arterial hypertension (PAH) are rare disorders characterised by occlusion of arterioles in the lung[1], leading to marked increases in pulmonary vascular resistance[2]. Life expectancy from diagnosis averages 3–5 years[3], with death ensuing from failure of the right ventricle.

Mutations in the gene encoding the bone morphogenetic protein type 2 receptor (BMPR2), a receptor for the transforming growth factor-beta (TGF-β) superfamily[4,5] account for over 80% of families with PAH, and approximately 20% of sporadic cases[6]. Mutations have been identified in genes encoding other components of the TGF-β/bone morphogenetic protein (BMP) signalling pathways, including ACVRL1[7] and ENG[8]. On endothelial cells, BMPR2 and ACVRL1 form a signalling complex, utilising ENG as a co-receptor. Case reports of rare sequence variation in the BMP signalling intermediaries, SMAD1, SMAD4 and SMAD9[9,10], provide compelling evidence for a central role of dysregulated BMP signalling in PAH pathogenesis.

Analysis of coding variation in BMPR2-negative kindreds revealed heterozygous mutations in genes not directly impacting on the TGF-β/BMP pathway, including CAV1[11], and the potassium channel, KCNK3[12]. Deletions and loss of function mutations in TBX4, an essential regulator of embryonic development, were identified in childhood-onset PAH[13]. A clinically and pathologically distinct form of PAH, known as pulmonary veno-occlusive disease or pulmonary capillary haemangiomatosis (PVOD/PCH), was shown recently to be caused by biallelic recessive mutations in EIF2AK4[14,15], a kinase in the integrated stress response.

The purpose of the present study was to identify additional rare sequence variation contributing to the genetic architecture of PAH, and to assess the relative contribution of rare variants in genes implicated in prior studies. A major finding is that rare likely causal heterozygous variants in several previously unidentified genes (ATP13A3, AQP1 and SOX17) were significantly overrepresented in the PAH cohort, and we provide independent validation for GDF2 as a causal gene.

## Results

**Description of the PAH cohort.** In total, 1048 PAH cases (1038 index cases and 10 related affected individuals) were recruited for WGS. Of these, 908 (86.7%) were diagnosed with idiopathic PAH, 58 (5.5%) gave a family history of PAH and 60 (5.7%) gave a history of drug exposure associated with PAH[16]. Twenty two cases (2.1%) held a clinical diagnosis of PVOD/PCH (Fig. 1a). Demographic and clinical characteristics of the PAH cohort are provided in Supplementary Table 1. An additional UK family was recruited separately for novel gene identification studies. Briefly, the proband was diagnosed at 12 years with a persistent ductus arteriosus and elevated pulmonary arterial pressure. Explant lung histology following heart-lung transplantation revealed the presence of plexiform lesions. Two of the proband's offspring were also diagnosed with childhood-onset PAH, one of which had an atrial septal defect. The proband's parents, siblings and a third child showed no evidence of cardiovascular disease.

**Pathogenic variants in previously reported PAH disease genes.** Our filtering strategy detected rare deleterious variation in previously reported PAH genes in 19.9% of the PAH cohort. For BMPR2, rare heterozygous mutations were identified in 160 of 1048 cases (15.3%). The frequency of BMPR2 mutations in familial PAH was 75.9%, in sporadic cases 12.2%, and 8.3% in anorexigen-exposed PAH cases. Forty-eight percent of BMPR2 mutations were reported previously[17], and the remainder were newly identified in this study. Fourteen percent of BMPR2 mutations resulted in the deletion of larger protein-coding regions ranging from 5 kb to 3.8 Mb in size. Supplementary Data 1 provides the breakdown of BMPR2 SNVs and indels, and the larger deletions are shown in Fig. 2a–c with a detailed summary in Supplementary Table 2.

Of the other genes previously reported in PAH we identified deleterious heterozygous rare variants in ACVRL1 (9 cases, 0.9%), ENG (6 cases, 0.6%), SMAD9 (4 cases, 0.4%), KCNK3 (4 cases, 0.4%), and TBX4 (14 cases, 1.3%). We identified one case with

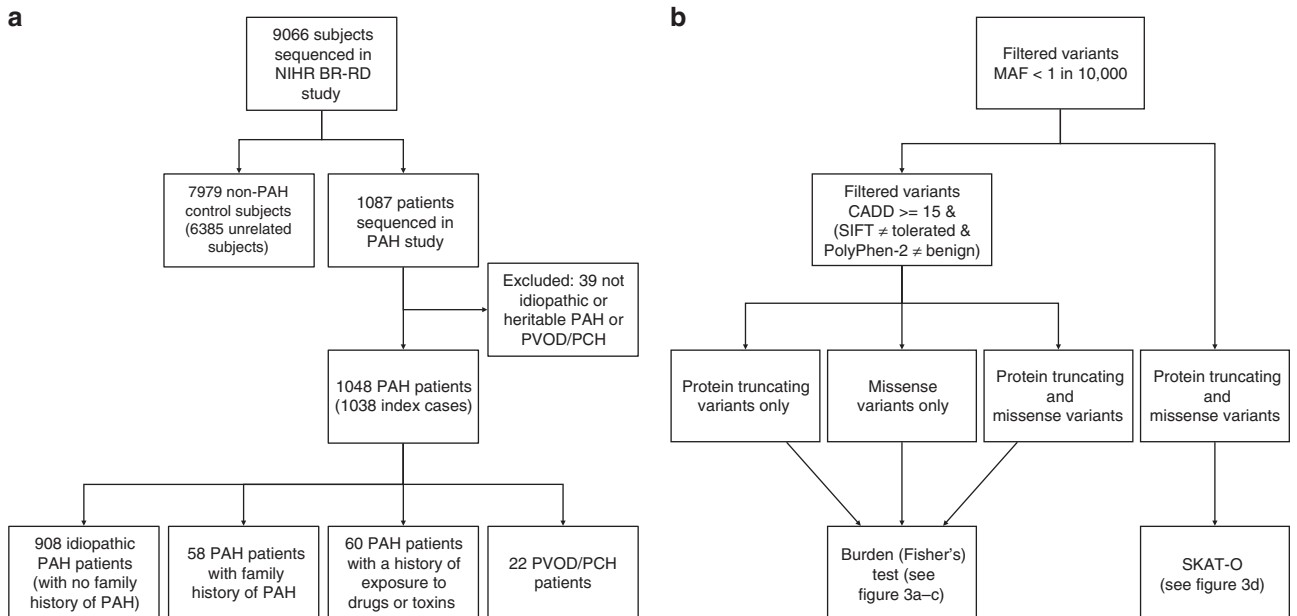

**Fig. 1** Flow diagrams illustrating **a** the composition of the NIHR BioResource—Rare Diseases (NIHR BR-RD) PAH study and **b** the analysis strategy to identify novel PAH disease genes. **a** The study comprised 1048 adult cases (aged 16 or over) attending specialist pulmonary hypertension centres from the UK (n = 731), and additional cases from France (n = 142), The Netherlands (n = 45), Germany (n = 82) and Italy (n = 48). **b** A series of case-control comparisons including and excluding cases with variants in previously reported disease genes were undertaken using complementary filtering strategies

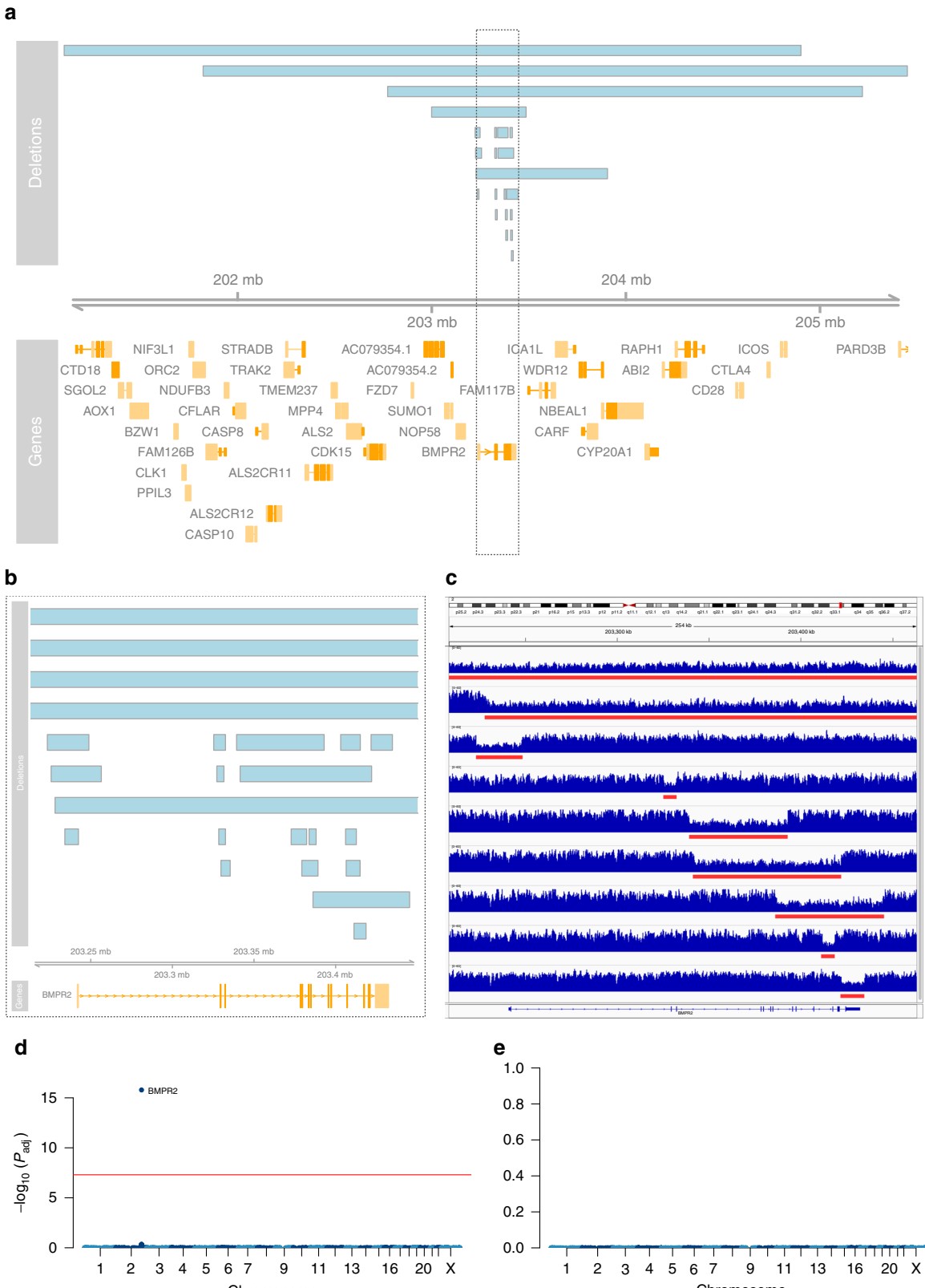

**Fig. 2** Analysis of copy number deletions. **a** Deletions affecting the *BMPR2* locus in 23 PAH cases. Genes are indicated in orange and labelled with their respective gene symbol. Deletions are drawn as blue boxes above the genome axis (grey) showing the genomic position on chromosome 2. The grey box highlights the location of *BMPR2*. **b** Locus zoom on *BMPR2* highlighting the focal deletions affecting one or more exons. **c** WGS coverage profiles of a selected set of smaller and larger deletions, visualised with the Integrative Genomics Viewer (IGV)[57], with deletions highlighted by red bars. **d** and **e** Manhattan plots of the genome-wide case-control comparison of large deletions. In **d**, all subject are considered. In **e**, subject with larger deletions affecting the *BMPR2* locus are excluded. The adjusted *P* value threshold of $5 \times 10^{-8}$ for genome-wide significance is indicated by the red line

highly deleterious variants in both *BMPR2* (p.Cys123Arg) and *SMAD9* (p.Arg294Ter). Details of consequence types, deleteriousness and conservation scores, and minor allele frequencies are provided in Supplementary Data 2. Fourteen cases (1.3%) with biallelic *EIF2AK4* mutations were found[18]. No pathogenic coding variants in *CAV1*, *SMAD1* or *SMAD4* were identified. Taken together, rare causal variation in non-*BMPR2* disease genes (*TBX4*, *ENG*, *ACVRL1*, *SMAD9*, *KCNK3* and *EIF2AK4*) accounted for 4.7% of the entire PAH cohort. The clinical characteristics of cases with variants in these previously reported genes are shown in Supplementary Table 3.

In a case-control comparison of the frequencies of deleterious variants confined to the previously reported PAH genes, we observed significant overrepresentation of rare variants in *BMPR2*, *TBX4*, *ACVRL1* and biallelic variants in *EIF2AK4* only ($P < 0.05$; Supplementary Table 4).

**Identification of novel PAH disease genes**. The strategy to identify novel causative genes in PAH employed a series of case-control analyses (Fig. 1b). To identify signals that might be masked by variants in previously reported PAH genes, we excluded subjects with rare variants and deletions in *BMPR2*, *EIF2AK4*, *ENG*, *ACVRL1*, *TBX4*, *SMAD9* and *KCNK3*. A genome-wide comparison of protein-truncating variants (PTVs), representative of high impact variants, identified a higher frequency of PTVs in *ATP13A3* (six cases) ($P_{adj} = 0.0346$). Moreover, we identified additional PTVs in several putative PAH genes, including *EVI5* (5 cases, 1 control) and *KDR* (4 cases, 0 controls; Fig. 3a), that require further validation to evaluate their contribution to PAH pathogenesis (Supplementary Table 5).

We next analysed rare missense variants overrepresented in the PAH cohort, again excluding subjects with variants in the previously reported PAH genes. This revealed significant overrepresentation of rare variants in *GDF2* after correction for multiple testing ($P_{adj} = 0.0023$), followed by *AQP1* (Fig. 3b and Supplementary Table 6). Next, in a combined analysis of rare missense variants and PTV, only *GDF2* remained significant ($P = 0.001$). Rare variants in additional putative genes occurred at higher frequency in cases compared to controls, including *AQP1*, *ALPPL2*, *ATP13A3*, *OR8U1*, *IFT74*, *FLNA*, *SOX17*, *ATP13A5*, *C3orf20* and *PIWIL1* (uncorrected $P$ value < 0.0005), but were not significant after correction for multiple testing (Fig. 3c and Supplementary Table 7).

In order to increase power to detect rare associations, we deployed SKAT-O on filtered rare PTVs and missense variants. Excluding previously reported genes, this analysis revealed an association with rare variants in *AQP1* ($P_{adj} = 4.28 \times 10^{-6}$) and *SOX17* ($P_{adj} = 6.7 \times 10^{-5}$; Fig. 3d). *AQP1* and *SOX17* were both also nominally significant in the combined burden tests, described above. Association was also found with rare variants in *MFRP* ($P_{adj} = 1.3 \times 10^{-5}$). However, we consider *MFRP* a false-positive finding for reasons given in the Discussion. Supplementary Table 8 shows the top 50 most significant genes identified by SKAT-O, providing further candidates to be evaluated in future studies. Details of rare variants in novel PAH genes (*GDF2*, *ATP13A3*, *AQP1*, *SOX17*) identified in cases are provided in Supplementary Data 3.

Notably, a genome-wide assessment of larger structural variation did not identify any additional large deletions after exclusion of subjects harbouring deletions in *BMPR2* (Fig. 2d, e).

The proportion of PAH cases with mutations in the new genes was 3.5%. The clinical characteristics of PAH cases with mutations in these genes are provided in Supplementary Table 3b.

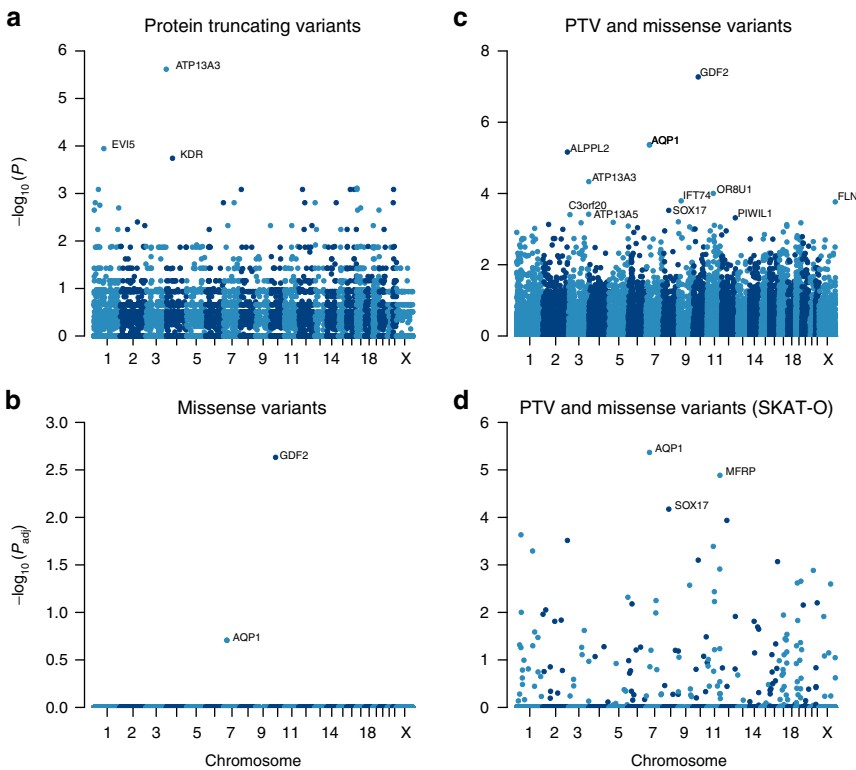

**Fig. 3** Manhattan plots of the rare variant analyses, having excluded cases carrying rare variants in previously established PAH genes. Filtered variants were grouped per gene. We tested for an excess of variants in PAH cases within genes using Fisher's exact test. The negative decadic logarithm of unadjusted or adjusted $P$-values are plotted against the chromosomal location of each gene. **a** Burden test of rare PTVs. **b** Burden test of rare deleterious missense variants. **c** Burden test combining rare PTVs and likely deleterious missense variants. **d** SKAT-O test of rare PTVs and missense variants

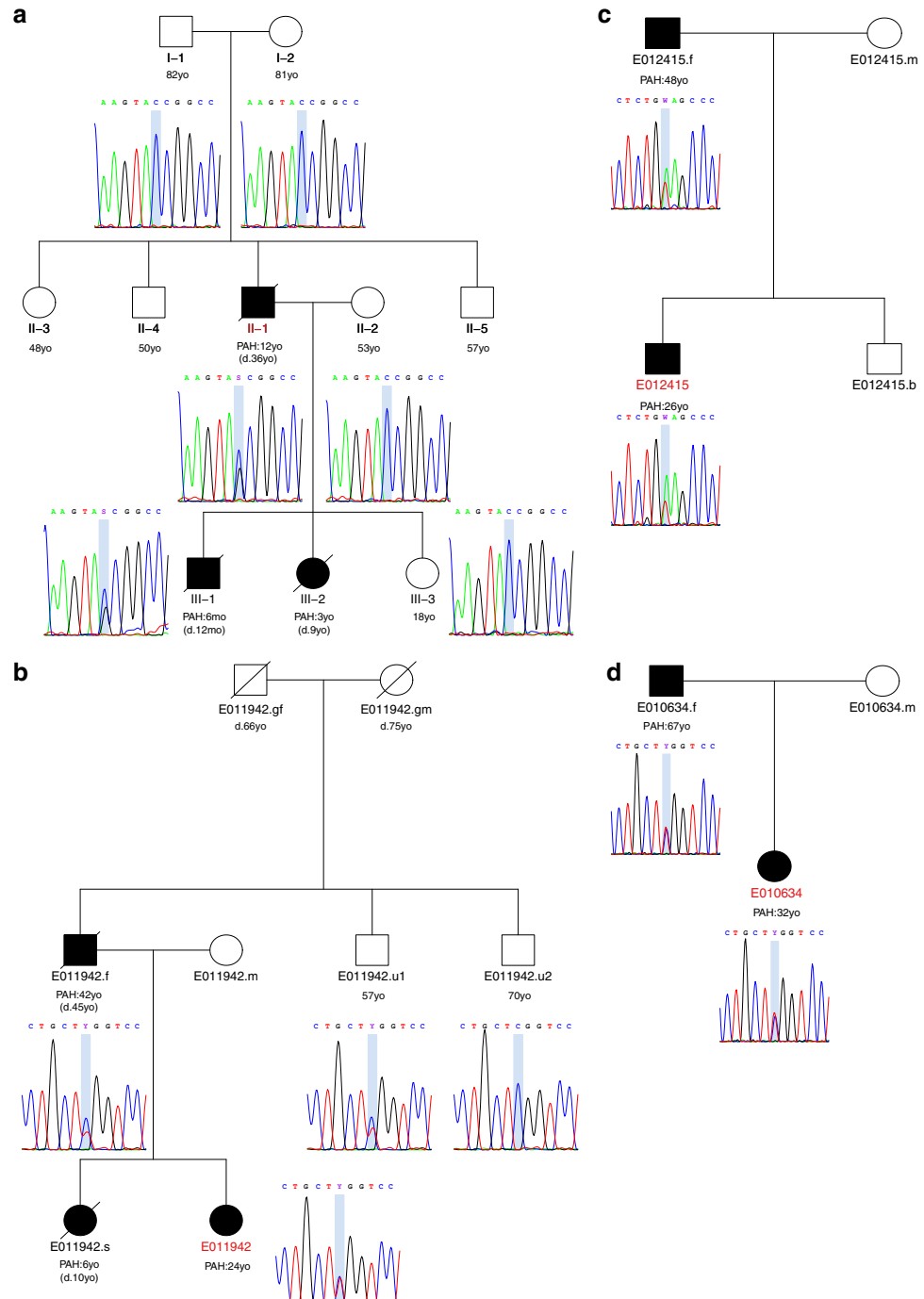

**Fig. 4** Pedigree structures and analysis of familial transmission of variants in *AQP1* and *SOX17*. **a** Individual II.1 harbours a heterozygous de novo *SOX17* c.411 C > G (p.Y137*) PTV resulting in a premature termination codon, which has been transmitted to the affected male (III.1). No unaffected family members carry the variant. No sample was available from subject III.2. **b** Proband E011942 has inherited a heterozygous *AQP1* c.583 C > T (p.R195W) missense variant from her affected father. No sample was available from the affected sister of the proband. The younger healthy uncle of the index case also carries the *AQP1* variant. No samples or further clinical information was available for the grandparents, who were not known to have cardiopulmonary disease. **c** Both the proband E012415 and her father are affected and carry the rare *AQP1* c.527 T > A (p.V176E) missense variant. There was no further information available about the siblings of the father. **d** Subject E010634 has inherited the heterozygous *AQP1* c.583 C > T (p.R195W) missense variant from her affected father. No rare variants in previously reported PAH genes were identified in any of theses families. Index cases are highlighted in red. d death, mo months old, yo years old

Of note, cases with mutations in *SOX17* and *AQP1* were significantly younger at diagnosis (32.8 ± 16.2 years ($P = 0.002$) and 36.9 ± 14.3 years ($P = 0.013$), respectively) compared to cases with no mutations in the previously established genes (51.7 ± 16.6 years).

**Non-coding variation around PAH disease genes**. An initial analysis for enrichment of variants in the non-coding sequence surrounding previously reported and newly identified PAH disease genes, including upstream gene regions, 5' UTRs, intronic sequence, 3' UTRs and downstream gene regions, did not detect

an significant overrepresentation in the PAH cohort. Details of the non-coding variants that passed the filtering strategy are provided in Supplementary Data 4.

**Independent validation and familial segregation analysis**. To provide further validation of the potentially causal role of mutations in the new genes identified, we examined whole-exome data from an independent UK family with three affected individuals across two generations. Microsatellite genotyping across chromosome 2q33 had previously demonstrated non-sharing of haplotypes in affected individuals, consistent with exclusion of linkage to the *BMPR2* locus. No pathogenic variants were identified in the protein-coding regions of the *BMPR2* gene or other TGF-β pathway genes. Analysis of exome sequence data from individual II-1 identified a novel heterozygous c.411 C > G (p. Y137*) PTV in the *SOX17* gene. Segregation analysis in the extended family demonstrated that the mutation had arisen de novo in the affected father (II-1) and was transmitted to the affected offspring (III-1). All unaffected family members were confirmed as wild-type (Fig. 4a).

Three HPAH subjects harbouring rare variants in *AQP1*, identified in the NIHR BR-RD WGS study, were also selected for familial co-segregation analysis (Fig. 4b–d). No pathogenic variants in any of the previously reported genes were identified in these families. The first pedigree comprised three affected individuals across two generations. Sanger sequencing confirmed the presence of the heterozygous *AQP1* c.583 C > T (p.R195W) missense variant in the proband (E011942), the affected father (E011942.f) and the healthy younger paternal uncle (E011942.u1). An additional unaffected uncle did not carry the AQP1 variant. These results indicate likely incomplete penetrance in the unaffected carrier, as observed in *BMPR2* families[19]. No additional clinical information was available for the deceased grandparents (Fig. 4b). The remaining two families comprised affected parent-offspring individuals. By Sanger sequencing we independently confirmed a heterozygous *AQP1* c.527 T > A (p. Val176Glu) missense variant in proband (E012415) and his affected father (Fig. 4c), as well as a heterozygous *AQP1* c.583 C > T (p.R195W) missense variant in proband (E010634) and her affected father (Fig. 4d). These results highlight recurrent *AQP1* variation across unrelated families and demonstrate co-segregation with the phenotype.

**Predicted functional impact of variants in novel PAH genes**. To evaluate the potential functional impact of rare variants identified in the likely causative new genes we performed structural analysis of *GDF2*, *ATP13A3*, *AQP1*, and *SOX17*. In addition we undertook a functional analysis of the *GDF2* variants identified.

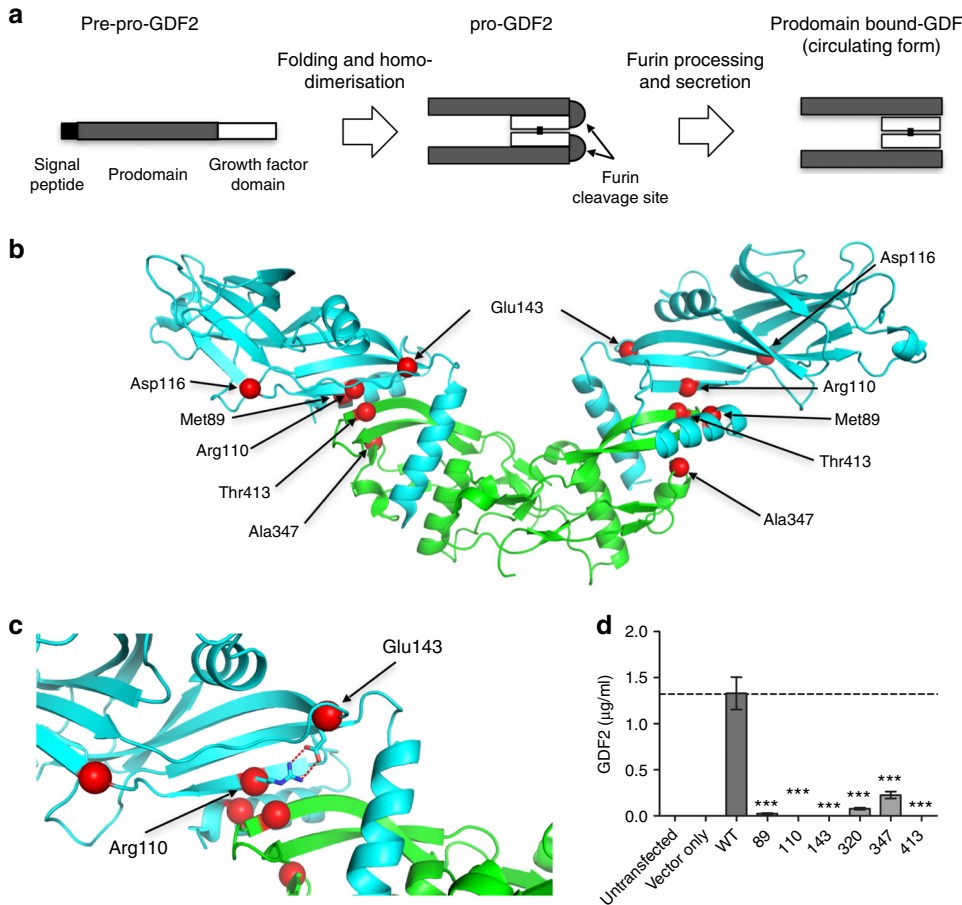

**Fig. 5** Structural analysis of *GDF2* mutations. **a** Schematic diagram of GDF2 processing. The pre-pro-protein is processed into the mature growth factor domain (GFD) bound to the prodomain upon secretion[61]. **b** Plot of *GDF2* mutations found only in PAH cases superimposed on the structure of prodomain bound GDF2 (PDB: 4YCG)[21]. The GDF2 growth factor domain is shown in green and the prodomain in cyan. **c** Magnified view of the Arg110 and Glu143 mutations. The wild-type amino acids make double salt bridges to stabilise the prodomain conformation at the interface between the growth factor domain and prodomain. The E143K and R110W mutations both disrupt these interactions, destabilising the interaction between the growth factor domain and prodomain. **d** GDF2 levels secreted into supernatants of HEK293T cells transfected with likely pathogenic variants found in PAH cases, compared with wild-type GDF2 and cells transfected with an empty vector. ***P < 0.001 by ANOVA

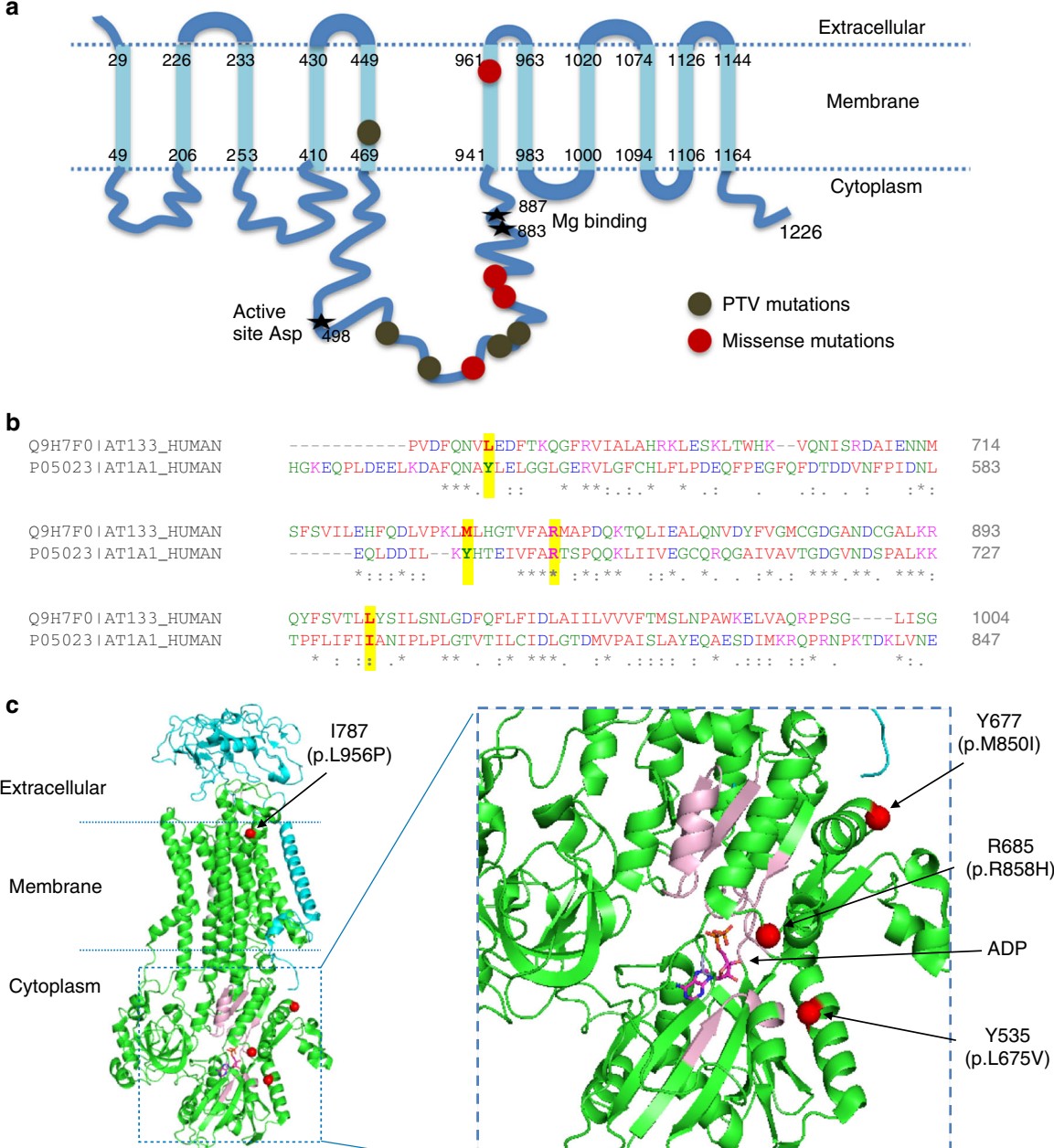

**Fig. 6** Structural analysis of *ATP13A3* mutations. **a** Topology of ATP13A3, plotted according to UniProtKB Q9H7F0. Frameshift and stop-gained mutations identified in PAH cases are shown as khaki circles, and missense mutations as red circles. Frameshift/stop-gained mutations are predicted to truncate the protein prior to the catalytic domain and essential Mg binding sites, leading to loss of ATPase activity. **b** Sequence alignment of ATP13A3 with ATP1A1 (P05024), of which the high resolution structure was used for the structural analysis in **c**. The conserved regions of ATP13A3 and ATP1A1, essential for ATPase activity[62], show good alignment (data not shown). Only regions containing the missense PAH mutations are shown, with positions of the four missense mutations highlighted in yellow above the sequences. **c** Structural analysis of the 4 PAH missense mutations plotted on the ATP1A1 crystal structure based on the sequence alignment in **b** (PDB: 3wgu)[63]. Green: α subunit (P05024), cyan: β subunit (P05027), grey: γ-subunit transcript variant a (Q58k79). Y535, Y677, R685 and I787 are the numbering in ATP1A1. Positions of the four missense mutations found in PAH are labelled and highlighted by red circles. **d** Magnified view of the cytoplasmic region of the ATPase, showing the presence of ADP at the active site. The conserved regions essential for ATPase activity are shown in light pink. The L675V and R858H mutations are located close to the ATP catalytic region

Heterozygous mutations in *GDF2* exclusive to PAH cases comprised 1 frameshift variant and 7 missense variants. *GDF2* encodes growth and differentiation factor 2, also known as bone morphogenetic protein 9 (BMP9), the major circulating ligand for the endothelial BMPR2/ACVRL1 receptor complex[20]. Amino acid substitutions were assessed against the published crystal structure[21] of the prodomain bound form of GDF2 (Fig. 5). Variants clustered at the interface between the prodomain and growth factor domain. Since the prodomain is important for the processing of GDF2, it is likely that amino acid substitutions reduce the stability of the prodomain-growth factor interface. In keeping with these predictions, HEK293T cells transfected with *GDF2* variants exclusive to PAH cases, demonstrated reduced secretion of mature GDF2 into the cell supernatants (Fig. 5d), compared with wild-type GDF2.

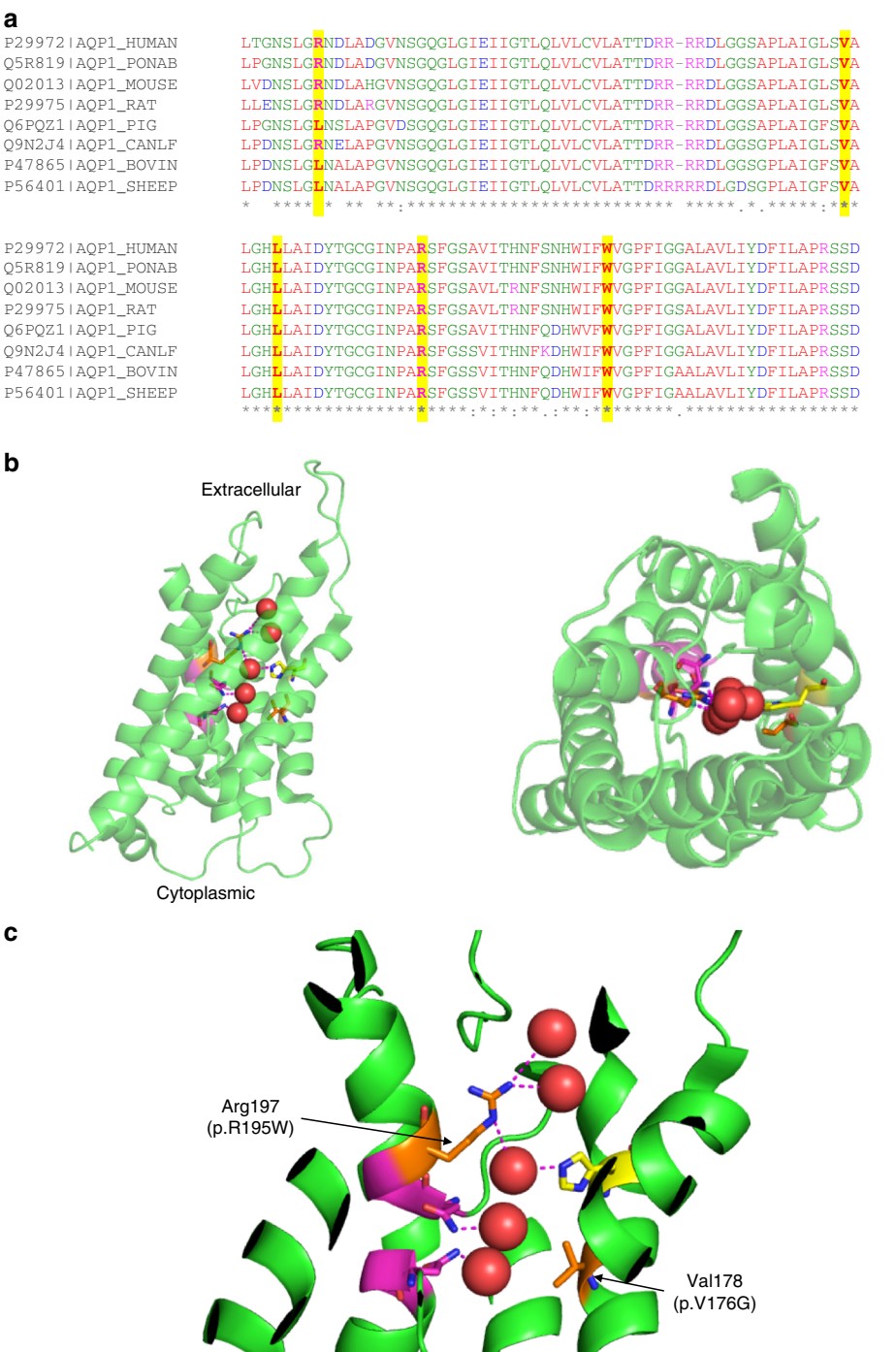

**Fig. 7** Structural analysis of *AQP1* mutations. **a** Multiple sequence alignment of human AQP1 with seven other mammals. The bovine AQP1 has the high resolution (2.2 Å) published structure. Mutations identified in PAH cases are highly conserved and highlighted in yellow. **b** Crystal structure of bovine AQP1 (PDB: 1j4n)[22]. Left: side view; right: top view from the extracellular direction. AQP1 is shown as a semi-transparent cartoon and five water molecules in the water channel are shown as red spheres. Key residues lining the water channels are represented with stick structures. **c** Magnified view of the water channel, with H-bonds connected to water molecules in the channel highlighted. Two asparagine-proline-alanine (NPA) motifs, essential for the water transporting function of AQP1, are shown in magenta. Conserved His180 that constricts the water channel is shown in yellow. Mutations found in PAH cases, Arg195Trp and Val176Glu, are labelled and shown as orange stick structures. Arg195 and His180 are highly conserved in the known water channels and are strong indicators of water channel specificity. Arg195Trp and Val176Glu mutations are predicted to disrupt the conformation of this conserved water channel

We identified three heterozygous frameshift variants, two stop gained, two splice region variants in *ATP13A3*, which are predicted to lead to loss of ATPase catalytic activity (Fig. 6a). In addition, we identified 4 heterozygous likely pathogenic missense variants in PAH cases, two near the conserved ATPase catalytic site and predicted to destabilise the conformation of the

catalytic domain (Fig. 6b–d). The distribution of variants (Fig. 6a) suggests that these mutations impact critically on the function of the protein.

The majority of rare variants identified in *AQP1*, which encodes aquaporin-1, are situated within the critical water channel (Fig. 7). In particular the p.Arg195Trp variant, identified

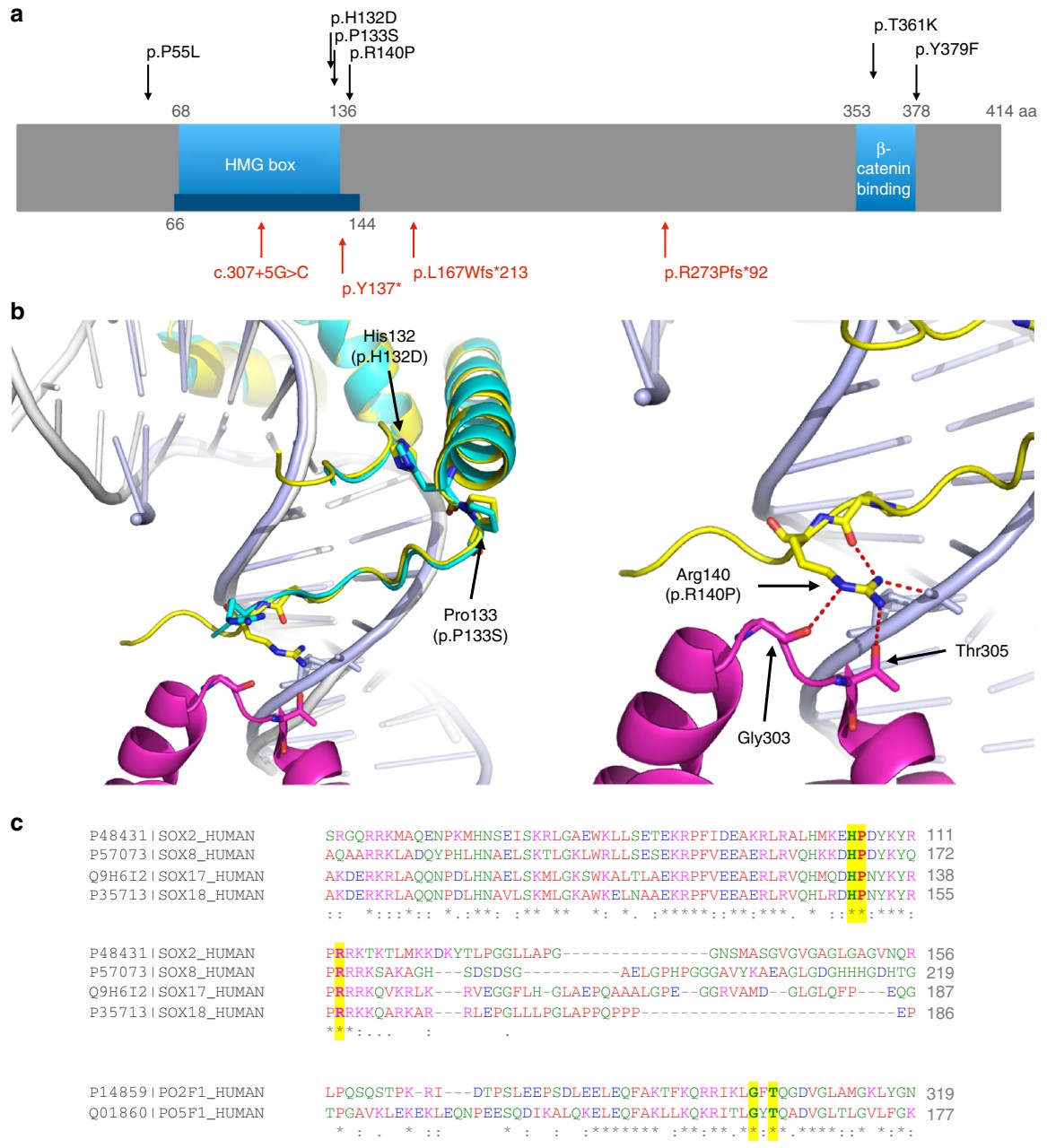

**Fig. 8** Structural analysis of *SOX17* mutations. **a** Schematic diagram of human SOX17 (Q9H6I2), based on UniProtKB annotation, and published reports[23]. Red arrows indicate PTVs and black arrows indicate missense mutations identified in PAH patients. The blue bar illustrates the region that is covered in the crystal structure (PDB: 3F27)[64]. The ability of SOX17 to activate transcription of target genes correlates with binding to β-catenin[23]. As illustrated, all PTVs lead to a loss of the β-catenin binding region. Two missense mutations are located within and very close to the minimum β-catenin binding regions, and both are highly conserved, indicating they are likely to be important for β-catenin binding. **b** Structural analysis of HMG domain missense mutations found in PAH patients. Left, Superposition of SOX17/DNA structure (Sox17: cyan, DNA: grey)[64] onto SOX2/DNA/Oct1 structure (PDB: 1GT0, Sox2: yellow, Oct1: magenta, DNA: light blue)[24]. Right: Magnified view of the interactions around Arg140 in the SOX2/DNA/Oct structure. Arg140 in SOX2 makes multiple H-bond interactions and mutating this Arg in SOX2 abolishes the interaction with transcription factors Pax6 and Oct4[24]. SOX2 and SOX17 both bind to Oct4[65] and SOX17 K122E mutant can replace SOX2 in maintaining stem cell pluripotency[65], indicating this region in SOX17 may interact with Oct4, similar to SOX2. The three missense mutations in *SOX17* will likely disrupt interaction with Oct4. **c** Supporting the analysis in **b**, sequence alignment shows that the HMG domain of SOX2 (P48431) and SOX17 as well as SOX8 (P57073) and SOX18 (P35713) share high sequence identity and the three mutations found in PAH (highlighted in yellow) are highly conserved emphasising their functional importance. Similarly, the Gly and Thr that interact with Arg140 in SOX2 (highlighted in yellow) are also conserved between Oct1 (PO2F1) and Oct4 (PO5F1)

in five PAH cases, locates at the hydrophilic face of the pore. This arginine at position 195 helps define the constriction region of the AQP1 pore structure and is conserved across the water-specific aquaporins[22]. Rare variants in *SOX17*, included four nonsense variants (including the PTV identified in the additional UK family) predicted to lead to loss of the beta-catenin binding region, and six missense variants predicted to disrupt interactions with Oct4 and beta-catenin[23,24] (Fig. 8).

GDF2 is known to be secreted from the liver, but the cellular localisation of proteins encoded by the other novel genes is less

well characterised. Thus we employed immunohistochemistry to examine localisation in the normal and hypertensive human pulmonary vasculature. Figure 9 shows that AQP1, ATP13A3 and SOX17 are predominantly localised to the pulmonary endothelium in normal human lung and to endothelial cells within plexiform lesions of patients with idiopathic PAH. In addition, we determined the relative mRNA expression levels of *AQP1*, *ATP13A3* and *SOX17* in primary cultures of pulmonary artery smooth muscle cells (PASMCs), pulmonary artery endothelial cells (PAECs) and blood outgrowth endothelial cells (BOECs)[25]. AQP1 was expressed in PASMCs and endothelial cells, with a trend towards higher levels in PASMCs (Fig. 10a). ATP13A3 was highly expressed in both cell types (Fig. 10b), whereas SOX17 was almost exclusively expressed in endothelial cells (Fig. 10c). Although AQP1 and SOX17 are known to play roles in endothelial function, the function of ATP13A3 in vascular cells is entirely unknown. Thus, we determined the impact of ATP13A3 knockdown on proliferation and apoptosis of BOECs. Loss of ATP13A3 led to marked inhibition of serum-stimulated proliferation of BOECs, and increased apoptosis in serum-deprived conditions (Fig. 10d–f).

## Discussion

We report a comprehensive analysis of rare genetic variation in a large cohort of index cases with idiopathic and heritable forms of PAH. Whilst we utilised WGS, the main goal was the identification of rare causal variation underlying PAH in the protein-coding sequence. The approach involved a rigorous case-control comparison using a tiered search for variants. First, we searched for high impact PTVs overrepresented in cases, having excluded previously established PAH genes. This revealed PTVs in *ATP13A3*, a poorly characterised P-type ATPase of the P5 subfamily[26]. There is little information regarding the function of the ATPase, *ATP13A3*, which appears widely expressed in mouse tissues[26]. Although, the precise substrate specificity is unknown, ATP13A3 plays a role in polyamine transport[27]. Based on available RNA sequencing data, *ATP13A3* is highly expressed in human pulmonary vascular cells and cardiac tissue (https://www.encodeproject.org). We confirmed that *ATP13A3* mRNA is expressed in primary cultured pulmonary artery smooth muscle cells and endothelial cells, and provide preliminary data that loss of *ATP13A3* inhibits proliferation and increases apoptosis of endothelial cells. These findings are consistent with the widely

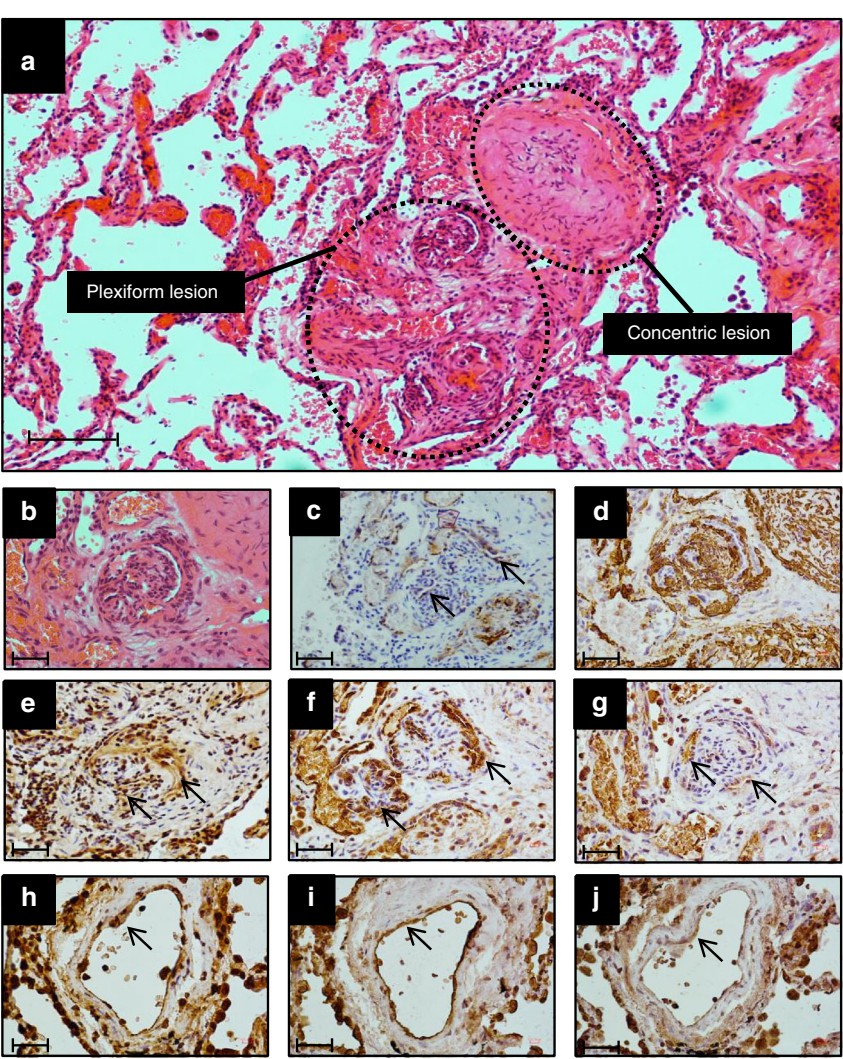

**Fig. 9** Immunolocalisation of AQP1, ATP13A3 and SOX17 in normal and PAH lung. The typical histological findings (haematoxylin and eosin staining) of concentric vascular lesions with associated plexiform lesions are shown (**a**). Higher magnification images of plexiform lesion (**b**), with frequent endothelialised channels (**c**; anti-CD31) surrounded by myofibroblasts (**d**; anti-SMα). Additional high magnification images demonstrating endothelial expression of ATP13A3 (**e**), AQP1 (**f**) and SOX17 (**g**) in PAH lung. Controls lung sections demonstrating predominantly endothelial expression of ATP13A3 (**h**), AQP1 (**i**) and SOX17 (**j**). (Scale bars = 50 μm)

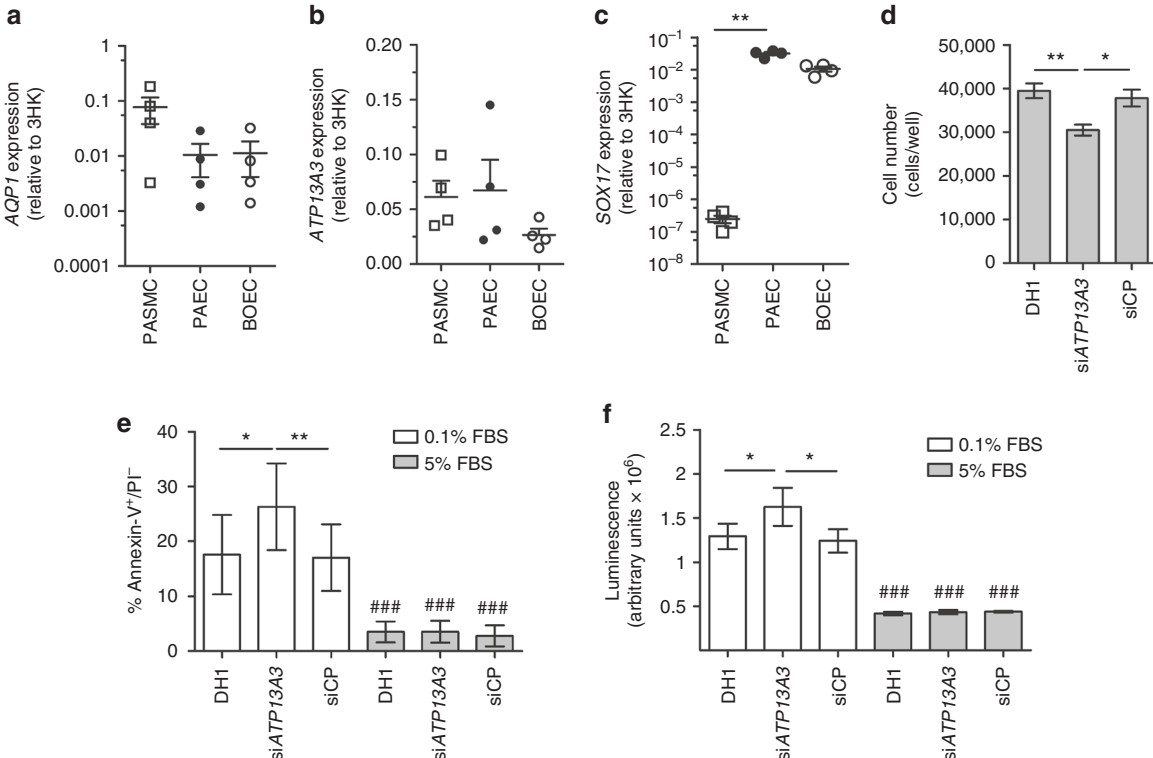

**Fig. 10** Functional studies of novel genes. **a–c** Expression of **a** *AQP1*, **b** *ATP13A3* and **c** *SOX17* mRNA in human pulmonary artery smooth muscle cells, pulmonary artery endothelial cells and blood outgrowth endothelial cells (BOECs) ($n = 4$ biological replicates of each). Relative expression of each transcript was normalised to three reference genes, *ACTB*, *B2M* and *HPRT*. **d** Proliferation of BOECs in 5% FBS over 6 days. Cells were transfected with DharmaFECT1 alone (DH1), si*ATP13A3* or non-targeting siRNA control (siCP) **e, f** Quantification of apoptosis in BOECs, defined as Annexin V+/PI− cells, in BOECs transfected with si*ATP13A3* or siCP in complex with DH1 followed by 24 h treatment with 0.1% FBS or 5% FBS ($n = 4$ biological repeats). **f** Measurement of apoptosis via Caspase-Glo 3/7 activity measurements in BOECs transfected with si*ATP13A3* or siCP in complex with DH1, followed by 16 h treatment in 0.1% FBS or 5% FBS. Data are from a single experiment ($n = 4$ wells) representative of 3 biological repeats. Data were analysed using a One-way analysis of variance with post hoc Tukey's test for multiple comparisons in **d** and **f**. Data were analysed using a repeated measures One-way analysis of variance with post hoc Tukey's for multiple comparisons in **e**. *$P < 0.05$, **$P < 0.01$ within treatment groups. ###$P < 0.001$ for effect of ligand against control for same transfection condition

accepted paradigm that endothelial apoptosis is a major trigger for the initiation of PAH[28,29]. It will be of considerable interest to determine the role of *ATP13A3* in vascular cells and whether it is functionally associated with BMP signalling, or represents a distinct therapeutic target in PAH.

Analysis of missense variation, and a combined analysis of all predicted deleterious variation, revealed that mutation at the *GDF2* gene is also significant determinant of predisposition to PAH. Of the new genes identified, *GDF2* provides further evidence for the central role of the BMP signalling pathway in PAH. *GDF2* encodes the major circulating ligand for the endothelial BMPR2/ACVRL1 receptor complex[20]. Taken together, the genetic findings suggest that a deficiency in GDF2/BMPR2/ACVRL1 signalling in pulmonary artery endothelial cells is critical in PAH pathobiology. The majority of *GDF2* variants detected in our adult-onset PAH cohort were heterozygous missense variants, in contrast to a previous case report of childhood-onset PAH due to a homozygous nonsense mutation[30]. The finding of causal *GDF2* variants in PAH cases, associated with reduced production of GDF2 from cells, provides further support for investigating replacement of this factor as a therapeutic strategy in PAH[31].

To maximise the assessment of rare variation in a case-control study design, we deployed the SKAT-O test. This approach revealed a significant association of rare variation in the aquaporin gene, *AQP1*, and the transcription factor encoded by *SOX17*. Of note, both *AQP1* and *SOX17* were within the top 8 ranked genes in our combined PTV and missense burden test analysis (Supplementary Table 7), providing further confidence in their causative contribution to PAH.

Aquaporin-1 belongs to a family of membrane channel proteins that facilitate water transport in response to osmotic gradients[22], and AQP1 is known to promote endothelial cell migration and angiogenesis[32]. Thus, approaches that maintain or restore pulmonary endothelial function could offer new therapeutic directions in PAH. Conversely, AQP1 inhibition in pulmonary artery smooth muscle cells ameliorated hypoxia-induced pulmonary hypertension in mice[33], suggesting that further studies are required to determine the key cell type impacted by *AQP1* mutations in human PAH, and the functional impact of these *AQP1* variants on water transport. The demonstration of familial segregation of *AQP1* variants with PAH provides further support for the potentially causal role of these mutations in disease. However, we also identified an unaffected *AQP1* variant carrier consistent with reduced penetrance, which is well described for other PAH genes, including *BMPR2*.

Although functional studies are required to confirm the mechanisms by which mutations in *SOX17* cause PAH, this finding provides additional support for the vascular endothelium as the major initiating cell type in this disorder. *SOX17* encodes the SRY-box containing transcription factor 17, which plays a fundamental role in angiogenesis[34] and arteriovenous

differentiation[35]. Moreover, conditional deletion of *SOX17* in mesenchymal progenitors leads to impaired formation of lung microvessels[36]. The demonstration of familial segregation of the *SOX17* p.Y137* PTV with early onset PAH provides additional evidence for a causal role for these variants in PAH. The co-existence of a patent ductus arteriosus in the index case and an atrial septal defect (ASD) in one of the affected offspring is of interest and suggests an association with congenital heart disease. Small ASDs are not uncommon in idiopathic PAH, and a more detailed clinical phenotyping of SOX17 mutation carriers will be required to determine whether the presence of ASDs and other congenital heart abnormalities are more common in carriers of these mutations.

Whilst the SKAT-O analysis also provided support for the *MFRP* gene, recessive biallelic mutations in *MFRP* cause retinal degeneration and posterior microphthalmos[37]. The expression of *MFRP* transcripts is largely confined to the central nervous system[38] and the majority of variants were present in the Genome Aggregation Database (GnomAD, http://gnomad.broadinstitute.org). On the basis of these considerations, variants in *MFRP* are unlikely to contribute to PAH aetiology.

This analysis provides new insights on the frequency and validity of previously reported genes in PAH. We confirmed that mutations in *BMPR2* are the most common genetic cause and validated rare causal variants in *ACVRL1*, *ENG*, *SMAD9*, *TBX4*, *KCNK3* and *EIF2AK4*. Although our findings question the validity of *CAV1*, *SMAD1* and *SMAD4* as causal genes, previous reports might represent private mutations occurring in very rare families. The use of WGS in this study allowed closer interrogation of larger deletions around the *BMPR2* locus than has been possible previously. Nevertheless, additional analyses are required to determine the full impact of structural variation (inversions, duplications, smaller deletions) at this and other loci.

The non-PAH cohort used in the case-control comparisons for this study comprised individuals, or relatives of individuals, with other rare diseases recruited to the NIHR BR-RD in the UK (Methods section). In general, for very rare causal variants, the comparison between PAH cases and non-PAH rare disease controls should not reduce our ability to detect over-representation of rare variants in a particular gene in the PAH cohort, if mutations in that gene are specific to PAH. However, if rare variants in a gene were responsible for more than one phenotype, it is possible that this would reduce the power to detect overrepresentation in the PAH cohort. For example, if mutations occurred in different functional domains of the expressed protein, this might lead to PAH if mutations affected one domain, but other phenotypes if they affected another domain. Overcoming this potential limitation will require additional analysis of the functional impact of variants and their distribution within a gene, and more detailed information on the phenotypes of subjects in the non-PAH group.

Taken together, this study identifies rare sequence variation in new genes underlying heritable forms of PAH, and provides a unique resource for future large-scale discovery efforts in this disorder. Mutations in previously established genes accounted for 19.9% of PAH cases. Including new genes identified in this study (*GDF2*, *ATP13A3*, *AQP1*, and *SOX17*), the total proportion of cases explained by mutations increased to 23.5%. It is likely that independent confirmation of the expanded list of putative genes identified in this study will increase further the proportion of cases explained by mutations, but this will require larger international collaborations. The results suggest that the genetic architecture of PAH, beyond mutations in *BMPR2*, is characterised by substantial genetic heterogeneity and consists of rare heterozygous coding region mutations shared by small numbers of cases. The contribution of rare variation within non-coding

regulatory regions to PAH aetiology remains to be determined. This will require functional annotation of regulatory and other non-coding regions specific for relevant cell types, further case-control analyses of these regions and ultimately functional studies of gene regulation to assess the pathogenicity of non-coding variants. Our findings to date provide support for a central role of the pulmonary vascular endothelium in disease pathogenesis, and suggest new mechanisms that could be exploited therapeutically in this life-limiting disease.

## Methods

**Ethics and patient selection.** Cases were recruited from the UK National Pulmonary Hypertension Centres, Universite Sud Paris (France), the VU University Medical Center Amsterdam (The Netherlands), the Universities of Gießen and Marburg (Germany), San Matteo Hospital, Pavia (Italy), and Medical University of Graz (Austria). All cases had a clinical diagnosis of idiopathic PAH, heritable PAH, drug-associated and toxin-associated PAH, or PVOD/PCH established by their expert centre. The non-PAH cohort for the case-control comparison comprised 6385 unrelated subjects recruited to the NIHR BR-RD study. All PAH and non-PAH patients provided written informed consent (UK Research Ethics Committee: 13/EE/0325), or local forms consenting to genetic testing in deceased patients and non-UK cases. An additional UK family diagnosed with HPAH was ascertained as described previously[39]. Blood and saliva samples were collected under written informed consent of the participants or their parents for use in gene identification studies (UK Research Ethics Committee: 08/H0802/32).

**Composition of non-PAH control cohort.** The non-PAH control cohort consisted of subjects with bleeding, thrombotic and platelet disorders (15.5%), cerebral small vessel disease (2.1%), Ehlers-Danlos syndrome (0.3%), subjects recruited to Genomics England Ltd (19.8%), hypertrophic cardiomyopathy (3.6%), intrahepatic cholestasis of pregnancy (4.1%), Leber hereditary optic neuropathy (0.9%), multiple primary tumours (7.8%), neuropathic pain disorder (2.6%), primary immune disorders (15.3%), primary membranoproliferative glomerulonephritis (2.3%), retinal dystrophies/paediatric neurology and metabolic disease (19.8%), stem cell and myeloid disorders (2.1%), steroid resistant nephrotic syndrome (3.6%), and others (0.3%), or their first degree relatives.

**High-throughput sequencing.** DNA extracted from venous blood underwent whole-genome sequencing using the Illumina TruSeq DNA PCR-Free Sample Preparation kit (Illumina Inc., San Diego, CA, USA) and Illumina HiSeq 2000 or HiSeq X sequencer, generating 100–150 bp reads with a minimum coverage of 15× for ~95% of the genome (mean coverage of 35×). Whole-exome sequencing was conducted for individual II-1 (Fig. 4a) using genomic DNA extracted from peripheral blood. Paired-end sequence reads were generated on an Illumina HiSeq 2000.

**Generation of analysis-ready data sets.** Sequencing reads were pre-processed by Illumina with Isaac Aligner and Variant Caller (v2, Illumina Inc.) using human genome assembly GRCh37 as reference. Variants were normalised, merged into multi-sample VCF files by chromosome using the gVCF aggregation tool agg (https://github.com/Illumina/agg) and annotated with Ensembl's Variant Effect Predictor (VEP). Following read alignment to the reference genome (GRCh37), variant calling and annotation of whole-exome data for individual II:1 were performed using GATK UnifiedGenotyper[40] and ANNOVAR[41], respectively. Annotations included minor allele frequencies from other control data sets (i.e. ExAC[42], 1000 Genomes Project[43] and UK10K[44]) as well as deleteriousness and conservation scores (i.e., CADD[45], SIFT[46], PolyPhen-2[47] and Gerp[48]) enabling further filtering and assessment of the likely pathogenicity of variants. To take forward only high quality calls, the pass frequency (proportion of samples containing alternate alleles that passed the original variant filtering) and call rate (proportion of samples with reference or alternate genotypes) were combined into the overall pass rate (OPR: pass frequency × call rate) and variants with an OPR of 80% or higher were retained.

**Estimation of ethnicity and relatedness.** We estimated the population structure and relatedness based on a representative set of SNPs using the R package GEN-ESIS to perform PC Air[49] and PC Relate[50], respectively. The selected 35,114 autosomal SNPs were present on Illumina genotyping arrays (HumanCoreExome-12v1.1, HumanCoreExome-24v1.0, HumanOmni2.5–8v1.1), do not overlap quality control excluded regions or multiallelic sites in the 1000 Genomes (1000 G) Phase 3 data set[43], do not have any missing genotypes in NIHR BR-RD, had a MAF of 0.3 or above and LD pruning was performed using PLINK[51] with a window size of 50 bp, window shift of 5 bp and a variance inflation factor threshold of 2. The 2,110 samples from the 1000 G Project including the European (EUR), African (AFR), South Asian (SAS) and East Asian (EAS) populations (excluding the admixed American population) were filtered for the selected SNPs and the filtered

data were used to perform a principal component analysis (PCA) using PC Air. We modelled the scores of the leading five principal components as data generated by a population specific multivariate Gaussian distribution and estimated the corresponding mean and covariance parameters. Genotypes from the NIHR BR-RD samples were projected onto the loadings for the leading five principal components from the 1000 G PCA and we computed the likelihood that each sample belonged to each subpopulation under a mixture of multivariate Gaussians models. Each sample was allocated to the population with the highest likelihood, unless the highest likelihood was similar to likelihood values for other populations, as might be expected for example under admixed ancestry or if the sample came from a population not included in 1000 G. Such ambiguous samples were labelled as "other". PC Relate was used to to identify related individuals in NIHR BR-RD. We used the first 20 PCs from PC Air to adjust for relatedness and extracted the pairwise Identity-By-State distances and kinship values. The pairwise information was used by Primus to infer family networks and calculate the maximum set of unrelated samples.

Of the 9110 NIHR BR-RD samples, we assigned 80.2% to Non-Finish European ($n = 7307$), 7.2% to South Asian ($n = 649$), 2.3% to African ($n = 213$), 0.08% to East Asian ($n = 78$), 0.02% to Finnish-European ($n = 19$) and 9.2% to Other ($n = 844$) and retrieved a maximum set of 7,493 unrelated individuals (UWGS10K), representing 82.2% of the entire NIHR BR-RD cohort.

**Cohort definition and allele frequency calculation**. Based on the relatedness analysis, we defined the following sample subsets: (a) the maximum number of unrelated non-PAH controls (UPAHC, $n = 6385$), (b) all affected PAH cases (PAHAFF, $n = 1048$), and (c) all unrelated PAH index cases (PAHIDX, $n = 1038$). These subsets were used to annotate the variants in the multi-sample VCF file with calculated minor allele frequencies using the fill-tags extension of BCFtools[52].

**Rare variant filtering**. Filtering of rare variants was performed as follows: (1) variants with a MAF less than 1 in 10,000 in UPAHC subjects, UK10K and ExAC were retained (adjusted for X chromosome variants to 1 in 8000); (2) variants with a combined annotation dependent depletion deleteriousness (CADD) score of less than 15 were excluded. CADD scores were calculated using the CADD web service (http://cadd.gs.washington.edu) for variants lacking a score; (3) premature truncating variants (PTVs) or missense variants of the canonical transcript were retained; 4) missense variants predicted to be both tolerated and benign by SIFT and PolyPhen-2, respectively, were removed.

To identify likely causative mutations (as reported in Supplementary Table 3), variants in previously reported and putative genes, identified in this study, were examined in more detail to exclude variants that did not segregate in families (where data available). Furthermore, variants shared between cases and non-PAH controls, as well as variants of uncertain significance that co-occurred with previously reported causative mutations or high impact PTVs were also excluded.

**Burden analysis of protein-truncating and missense variants**. Filtered variants were grouped per gene and consequence type (predicted PTV/missense) and subjects with at least one variant were counted (no double counting) per group and tested for association with disease. We applied a one-tailed Fisher's exact test with post hoc Bonferroni correction to calculate the $P$ value for genome-wide significance.

**Rare variant analysis using SKAT-O**. To further investigate the aggregated effect that rare variants contribute to PAH aetiology, we applied a Sequence Kernel Association test (SKAT-O). SKAT-O increases the power of discovery under different inheritance models by combining variance-component and burden tests. Variants were filtered based on MAF as specified above, and only PTV and missense variants were included. For the analysis we implemented SKAT-O in RvTests v1.9.9[53] with default parameters and weights being Beta(1,25), and applying a correction for read length, gender and the first five principal components of the ethnicity PCA. Variants were collapsed considering only the protein-coding region in the canonical transcript of the protein-coding genes in the genome assembly GRCh37.

**Analysis of large deletions**. Copy number variation was identified using Canvas[54] and Manta[55]. Deletions called by both Manta and Canvas with a reciprocal overlap of ≥20% were retained. Of these, deletions were excluded if both failed standard Illumina quality metrics or overlapped with known benign deletions in healthy cohorts[56]. Deletions with a reciprocal overlap of ≥50% between samples were merged and filtered for a frequency of less than 1 in 1000 in WGS10K and overlapping exonic regions of protein-coding genes (GRCh37 genome assembly). The number of subjects with deletions were added up by gene (no double counting of subjects) and tested for association with the disease. We applied a one-tailed (greater) Fisher's exact test with Bonferroni post hoc correction for multiple testing to determine the $P$ values for genome-wide significance.

**Confirmation of variants**. Variant sequencing reads for SNVs, indels and deletions were visualised for validation on Integrative Genomes Viewer (IGV)[57], and were confirmed by diagnostic capture-based high-throughput sequencing, if the IGV inspection was not satisfactory. For the familial segregation analysis, linkage to the *BMPR2* locus was first examined by microsatellite genotyping analysis. Mutation screening of the *BMPR2*, *ACVRL1*, *ENG*, *AQP1* and *SOX17* genes was conducted by capillary sequencing using BigDye Terminator v3.1 chemistry. All DNA fragments were resolved on an ABI Fragment Analyzer (Applied Biosystems). All primer sequences are listed in Supplementary Table 9. The family trees were drawn using the R package FamAgg[58].

**Structural analysis of novel variants**. The domain structures and the functional groups of the novel *PAH* genes were plotted according to the entry in UniProtKB. Clustal Omega was used for sequence alignment. Structural data were obtained from RCSB Protein Data Bank and analysed according to published reports. Figures were generated using PyMOL Molecular Graphics System.

**Production of pGDF2 wild type and variant proteins**. The cloning of human wild-type pro-GDF2 (pGDF2) in pCEP4 has been described previously[59]. Site-directed mutagenesis was performed according to the manufacturer's instructions (QuickChange Site-Directed Mutagenesis Kit, Agilent Technologies). Mutations were confirmed by Sanger sequencing. HEK-EBNA cells were transfected with plasmids containing either wild-type or mutant pGDF2 for 14 h. The transfecting supernatant was removed and replaced with CDCHO media (Invitrogen) for 5 days to express the proteins. The conditioned media containing GDF2 and the variants were collected and snap-frozen on dry-ice before being stored at −80 °C. For each variant, conditioned media from three independent transfections were collected for further characterisation.

**GDF2 ELISA**. High binding 96-well ELISA plates (Greiner, South Lanarkshire, UK) were coated with 0.2 μg/well of mouse monoclonal anti-human GDF2 antibody (R&D Systems, Oxfordshire, UK) in PBS (0.1 M phosphate pH7.4, 0.137 M NaCl, 2.7 mM KCl, Sigma) overnight at 4 °C in a humidified chamber. Plates were washed with PBS containing 0.05% (v/v) Tween-20 (PBS-T), followed by blocking with 1% bovine serum albumin in PBS-T (1% BSA/PBS-T) for 90 min at room temperature. Recombinant human GDF2 standards (1–3000 pg/ml) or conditioned media samples (100 μl/well of 1:30, 1:100, 1:300, 1:1000, 1:3000 and 1:10,000 dilutions) were then added and incubated for 2 h at room temperature. After washing, plates were then incubated with 0.04 μg/well biotinylated goat anti-human GDF2 (R&D Systems) in 1% BSA/PBS-T for 2 h. Plates were washed, then incubated with ExtrAvidin(r)-Alkaline phosphatase (Sigma) diluted 1:400 in 1% BSA/PBS-T for 90 min. Plates were washed with PBS-T followed by water. The ELISA was developed with a colorimetric substrate comprising 1 mg/ml 4-Nitrophenyl phosphate disodium salt hexahydrate (Sigma) in 1 M Diethanolamine, pH 9.8 containing 0.5 mM MgCl$_2$. The assay was developed in the dark at room temperature and the absorbance measured at 405 nm.

**Cell culture and treatments**. Distal human pulmonary artery smooth muscle cells (PASMCs) were cultured from explants dissected from lung resection specimens. Small pulmonary arterioles (0.5 to 2 mm diameter) were dissected and divided into small pieces before plating in T25 flasks. Explants were left to adhere for 2 h and then incubated in DMEM/20% FBS plus amino acids at 37 °C in 95% air/5% CO2 until PASMCs had formed confluent monolayers. Cells were then trypsinized, and for subsequent passages cells were maintained in DMEM supplemented with 10% FBS. The cellular phenotype of PASMCs was confirmed by positive immunofluorescence staining with anti-smooth muscle specific alpha-actin (Clone IA4 Sigma-Aldrich; 1:100 dilution). The derivation of human tissues and cells was approved by Papworth Hospital ethical review committee (Ref 08/H0304/56 + 5) and all subjects provided informed and written consent.

Human blood outgrowth endothelial cells (BOECs) were derived from 40–80 ml of peripheral venous blood isolated from healthy subjects. The study was approved by the Cambridgeshire 3 Research Ethics Committee (Ref 11/EE/0297), and all subjects provided informed and written consent. BOECs were cultured in 10% FBS supplemented with EGM-2MV (Life Technologies, Carlsbad, CA). Cells were used between passages 4 and 8[60]. The endothelial phenotype of BOECs was determined by flow cytometry for expression of endothelial surface markers, as described previously[25]. Cells were routinely tested to exclude mycoplasma infection.

Human pulmonary artery endothelial cells (PAECs) were purchased from Lonza (Cat. No. CC-2530; Basel, Switzerland). Cells were maintained in EGM-2 with 2% FBS (Lonza). PAECs were used for experiments between passages 4 and 8. For experiments cells were cultured in the presence of EBM-2 containing Antibiotic-Antimycotic (Invitrogen, Renfrewshire, UK). Cells were routinely tested to exclude mycoplasma contamination.

**RNA preparation and quantitative reverse transcription-PCR**. Total RNA was extracted using RNeasy Mini Kit with DNAse digestion (Qiagen, West Sussex, UK), according to the manufacturer's instructions. cDNA was prepared from 1 μg of

RNA using High Capacity Reverse Transcriptase kit (Applied Biosystems, Foster City, CA). Quantitative PCR reactions employed MicroAmp optical 96-well reaction plates (Applied Biosystems). 50 ng µl$^{-1}$ cDNA was used with SYBR Green Jumpstart Taq Readymix (Sigma-Aldrich), ROX reference dye (Invitrogen) using custom made sense and anti-sense primers (all 200 nmol l$^{-1}$). Primers for human ACTB (encoding β-actin), AQP1, ATP13A3, B2M, HPRT and SOX17 were designed using PrimerBLAST (https://www.ncbi.nlm.nih.gov/tools/primer-blast/; Supplementary Table 9). Reactions were amplified on a Quantstudio 6 Real-Time PCR system (Applied Biosystems). The relative abundance of each target gene in different cell lines was compared using the equation $2^{-(CtGOI-Ct3HK)}$, where Ct3HK corresponded to the arithmetic mean of the Cts for ACTB, B2M and HPRT for each sample. For expression analysis of siRNA knockdown, the $2^{-(\Delta\Delta Ct)}$ method was used and fold expression determined relative to the DH1 control.

**siRNA transfection**. Prior to transfection, cells were preincubated in Opti-MEM-I reduced serum media (Invitrogen) for 2 h before transfection with 10 nM siRNA that had been lipoplexed for 20 min at RT with DharmaFECT1 (GE Dharmacon, Lafayette, CO). Cells were then incubated with the siRNA/DharmaFECT1 complexes for 4 h at 37 °C before replaced by full growth media. Cells were kept in growth media for 24 h before further treatment. Knockdown efficiency was confirmed by mRNA expression or immunoblotting. For proliferation assays, parallel RNA samples were collected both on day0 and day6, confirming that ATP13A3 expression was reduced by >90% on Day 0 and still reduced by >70% at Day 6. For all other assays, parallel RNA samples were collected on the day of the experiment to confirm knockdown, which was >90%. The siRNAs used were oligos targeting ATP13A3 (SASI_Hs02_00356805) from Sigma-Aldrich and ON-TARGET plus non-targeting Pool (siCP; GE Dharmacon).

**Flow cytometric apoptosis assay**. BOECs were plated 150,000/well into 6-well plates and transfected with siATP13A3 or siCP lipoplexed with DharmaFECT1. Cells were then serum-starved in EBM-2 (Lonza) containing 0.1% FBS and A/A for 8 h before treating with EBM-2 and A/A containing either 0.1%FBS or 5% FBS for another 24 h. Cells were then trypsinized and after washing with PBS, stained using the FITC Annexin V Apoptosis Detection Kit I (BD Biosciences). For each condition, dual-staining of 5 µl FITC conjugated Annexin V and 5 µl propidium iodide (PI) were added and incubated at room temperature for 15 min. For the single staining controls for compensation, either 5 µl FITC Annexin V or 5 µl PI was added into non-transfected cells. All samples were analysed on BD Accuri™ C6 Plus platform (BD Biosciences). Data were collected and analysed using FlowJo software, with AnnexinV$^+$/PI$^-$ cells defined as early apoptotic (Treestar).

**Caspase-Glo 3/7 assay**. BOECs were seeded at a density of 150,000/well into 6-well plates and transfected with siATP13A3 or siCP lipoplexed with Dharma-FECT1. For each condition, cells were trypsinized from 6-well plates and reseeded in triplicates into a 96-well plate at a density of 15,000–20,000/well and left to adhere overnight. Cells were quiesced in EBM-2 containing 0.1% FBS for 24 h before treating with or without EBM-2 and A/A containing either 0.1% FBS or 5% FBS for 16 h. For measuring caspase activities, 100ul Caspase-Glo® 3/7 Reagent (G8091 Promega) was added into each well, incubated and mixed on a plate shaker in the dark for 30 min at room temperature. The lysates were transferred to a white-walled 96-well plate and luminescence was read in a GloMax® luminometer (Promega).

**Data availability**. WGS data of PAH cases included in this manuscript and eligible for public release according to the UK Research Ethics rules have been deposited in the European Genome-phenome Archive (EGA) at the EMBL—European Bioinformatics Institute under accession number EGAD00001003423.

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

## Acknowledgements

The UK National Institute for Health Research BioResource—Rare Diseases (NIHR BR-RD) and the BHF/MRC UK National Cohort of Idiopathic and Heritable PAH made this study possible. We gratefully acknowledge the participation of patients recruited to the NIHR BR-RD. We thank the NIHR BR-RD staff and co-ordination teams at the University of Cambridge, and the research nurses and coordinators at the specialist pulmonary hypertension centres involved in this study. The UK National Cohort of Idiopathic and Heritable PAH is supported by the NIHR, the British Heart Foundation (BHF) (SP/12/12/29836), the BHF Cambridge Centre of Cardiovascular Research Excellence, the UK Medical Research Council (MR/K020919/1), the Dinosaur Trust, BHF Programme grants to R.C.T. (RG/08/006/25302) and N.W.M. (RG/13/4/30107), and the UK NIHR Cambridge Biomedical Research Centre. Funding for whole-exome sequencing was provided through a Bart's Charity award (MGU0205) to R.C.T. and D.v. H. N.W.M. is a BHF Professor and NIHR Senior Investigator. CH is a NIHR Rare Disease Translational Research Collaboration Clinical PhD Fellow. L.S. is supported by the Wellcome Trust Institutional Strategic Support Fund (204809/Z/16/Z) awarded to St. George's, University of London. C.J.R. is supported by a BHF Intermediate Basic Science Research Fellowship (FS/15/59/31839). A.L. is supported by a BHF Senior Basic Science Research Fellowship (FS/13/48/30453). We acknowledge the support of the Imperial NIHR Clinical Research Facility, the Netherlands CardioVascular Research Initiative, the Dutch Heart Foundation, Dutch Federation of University Medical Centres, the Netherlands Organisation for Health Research and Development and the Royal Netherlands Academy of Sciences. We also gratefully acknowledge Dr Claudia Cabrera in the NIHR Barts Cardiovascular Biomedical Research Centre for bioinformatics support. We thank all the patients and their families who contributed to this research and the Pulmonary Hypertension Association (UK) for their support.

## Author contributions

S.G., N.W.M. and W.H.O. conceived and designed the research. S.G., M.H, M.B. and C.H. processed the data and performed the statistical analysis. S.G., M.H, M.B., C.H. and N.W.M. drafted the manuscript. L.S., R.D.M. and R.C.T. conducted the *SOX17* familial segregation analyses. W.L. performed the structural analysis of the rare variants. R.S. generated the mutant cells. J.H., R.M.S, B.L. and P.D.U. conducted the functional experiments on the novel disease genes. M.S. performed the immunohistochemistry for novel gene products. L.C.D. helped with the assessment of pertinent findings. O.S. was involved with data analysis. D.W. participated in DNA extraction, sample QC and plating. L.S., R.D.M., S.H., M.A., C.J.R., W.H.O., N.S., A.L., R.C.T. and M.R.W. helped with data analysis and interpretation and made critical revision of the manuscript for important intellectual content. J.M.M., C.M.T. and K.Y. coordinated data collection. N.W.M. and W.H.O. handled the funding for the study. All other authors were responsible for data acquisition and recruitment of subjects to the study and helped to draft the final version of the manuscript.

## Additional information

**Competing interests:** The authors declare no competing interests.

Stefan Gräf [1,2,3], Matthias Haimel [1,2,3], Marta Bleda [1], Charaka Hadinnapola [1], Laura Southgate [4,5],
Wei Li[1], Joshua Hodgson[1], Bin Liu[1], Richard M. Salmon[1], Mark Southwood[6], Rajiv D. Machado[7],
Jennifer M. Martin [1,2,3], Carmen M. Treacy[1,6], Katherine Yates[1,2,3], Louise C. Daugherty[2,3],
Olga Shamardina [2,3], Deborah Whitehorn[2,3], Simon Holden[8], Micheala Aldred [9], Harm J. Bogaard[10],
Colin Church[11], Gerry Coghlan[12], Robin Condliffe[13], Paul A. Corris[14], Cesare Danesino[15,16], Mélanie Eyries[17],
Henning Gall[18], Stefano Ghio[16], Hossein-Ardeschir Ghofrani[18,19], J. Simon R. Gibbs[20], Barbara Girerd[21],
Arjan C. Houweling[10], Luke Howard[19], Marc Humbert[21], David G. Kiely[13], Gabor Kovacs[22,23],
Robert V. MacKenzie Ross[24], Shahin Moledina[25], David Montani [21], Michael Newnham[1], Andrea Olschewski[22],
Horst Olschewski[22,23], Andrew J. Peacock[11], Joanna Pepke-Zaba[6], Inga Prokopenko [19],
Christopher J. Rhodes [19], Laura Scelsi[16], Werner Seeger[18], Florent Soubrier[17], Dan F. Stein [1],
Jay Suntharalingam[24], Emilia M. Swietlik[1], Mark R. Toshner [1], David A. van Heel [26],
Anton Vonk Noordegraaf[10], Quinten Waisfisz[10], John Wharton [19], Stephen J. Wort[27,19],
Willem H. Ouwehand[2,3], Nicole Soranzo [2,28], Allan Lawrie [29], Paul D. Upton [1], Martin R. Wilkins[19],
Richard C. Trembath [5] & Nicholas W. Morrell [1,3]

[1]Department of Medicine, University of Cambridge, Cambridge CB2 0QQ, United Kingdom. [2]Department of Haematology, University of Cambridge, Cambridge CB2 0PT, United Kingdom. [3]NIHR BioResource—Rare Diseases, Cambridge CB2 0PT, United Kingdom. [4]Molecular and Clinical Sciences Research Institute, St George's, University of London, London SW17 0RE, United Kingdom. [5]Division of Genetics & Molecular Medicine, King's College London, London WC2R 2LS, United Kingdom. [6]Royal Papworth Hospital, Papworth Everard, Cambridge CB23 3RE, United Kingdom. [7]Institute of Medical and Biomedical Education, St George's University of London, London SW17 0RE, United Kingdom. [8]Addenbrooke's Hospital, Cambridge CB2 0QQ, United Kingdom. [9]Cleveland Clinic, Cleveland, Ohio 44195, United States. [10]VU University Medical Center, Amsterdam 1007 MB, The Netherlands. [11]Golden Jubilee National Hospital, Glasgow G81 4DY, United Kingdom. [12]Royal Free Hospital, London NW3 2QG, United Kingdom. [13]Sheffield Pulmonary Vascular Disease Unit, Royal Hallamshire Hospital, Sheffield S10 2JF, United Kingdom. [14]University of Newcastle, Newcastle NE1 7RU, United Kingdom. [15]Department of Molecular Medicine, University of Pavia, Pavia 27100, Italy. [16]Fondazione IRCCS Policlinico San Matteo, Pavia 27100, Italy. [17]Département de génétique, hôpital Pitié-Salpêtrière, Assistance Publique-Hôpitaux de Paris, and UMR_S 1166-ICAN, INSERM, UPMC Sorbonne Universités, Paris 75252, France. [18]University of Giessen and Marburg Lung Center (UGMLC), member of the German Center for Lung Research (DZL) and of the Excellence Cluster Cardio-Pulmonary System (ECCCPS), Giessen 35392, Germany. [19]Imperial College London, London SW7 2AZ, United Kingdom. [20]National Heart & Lung Institute, Imperial College London, London SW3 6LY, United Kingdom. [21]Université Paris-Sud, Faculté de Médecine, Université Paris-Saclay; AP-HP, Service de Pneumologie, Centre de référence de l'hypertension pulmonaire; INSERM UMR_S 999, Hôpital Bicêtre, Le Kremlin-Bicêtre, Paris 94270, France. [22]Ludwig Boltzmann Institute for Lung Vascular Research, Graz 8010, Austria. [23]Medical University of Graz, Graz 8036, Austria. [24]Royal United Hospitals Bath NHS Foundation Trust, Bath BA1 3NG, United Kingdom. [25]Great Ormond Street Hospital, London WC1N 3JH, United Kingdom. [26]Blizard Institute, Queen Mary University of London, London E1 2AT, United Kingdom. [27]Royal Brompton Hospital, London SW3 6NP, United Kingdom. [28]Wellcome Trust Sanger Institute, Hinxton CB10 1SA, United Kingdom. [29]Department of Infection, Immunity & Cardiovascular Disease, University of Sheffield, Sheffield S10 2RX, United Kingdom. These authors jointly supervised this work: Stefan Gräf, Nicholas W. Morrell. These authors contributed equally: Stefan Gräf, Matthias Haimel, Marta Bleda, Charaka Hadinnapola, Laura Southgate.

