## [Peer Review File(PDF 193 kb) · Nature Communications]

Reviewers' comments:

Reviewer #1 (Remarks to the Author):

In this study, Graf S et al. investigated whether rare variants of newly identified genes were found in a population of patients with various PAH types, including sporadic PAH, heritable PAH, and PVD (a rare form of PAH). They sought variants by whole genome sequencing in 1038 index patients with PAH and 6385 controls with other rare diseases. The results substantiate previous findings on the frequency of BMPRII mutations in the various subgroups. In addition, 4.7% of the cohort with PAH had rare causal variations in non-BMPRII genes including TBX4, ENG, ACVRL1, SMAD9, KCNK3, and EIF2AK4.

Finally, 6 patients carried protein-truncating variants (PTVs) in ATP13A3. Excluding previously reported genes, an association was evident with rare variants in AQP1 and SOX17. An analysis of rare missense mutations revealed overrepresentation of rare variants in GDF2 (encoding BMP9), and transfecting cultured cells with these variants decreased the release of BMP9.

These data highlight associations linking common and less common gene variants to PAH. The study deserves credit for providing a comprehensive picture of genes associated with PAH in a mixed population of patients with heritable PAH, non-heritable PAH, and PVD. The results, however, induce some frustration, as the number of genes newly identified as associated with PAH is small, their frequency low, and their causal effect unproven.

One possible explanation to the low yield of the study is that the controls were patients with rare diseases other than PAH. This point is discussed at the end of the discussion. It probably limited the power of the study. The same methodology with a different control population may well have produced different results. Are the authors planning such a study to assess this hypothesis?

In the patients with PAH, both the nature and the frequency of the pathogenic variants are consistent with earlier reports. However, the large study population probably has considerable overlap with populations included in previous studies. Thus, it is unsurprising that the genes and their frequencies are similar to those in reported previously. The authors should indicate which patients were included in earlier studies.

Some previously reported gene mutations, such as Alk1, were not identified in this study.

The term "causative genes" used in the title is an overstatement. Whether the small number of uncommon variants identified in this study have causal effects is unproven.

An issue worthy of investigation is whether the newly identified genes are associated with the previously known genes. For example, are some of these novel genes associated with the BMPRII mutation and suspected to increase penetrance?

A few experiments have been done to localize AQP1 and SOX17, and a few studies have assessed the function of ATP13A3. However, the function of these genes in PAH remains unclear.

Reviewer #2 (Remarks to the Author):

In this report Graf et al. present data from a genomic study of a large cohort of patients with IPAH. They used WGS to sequence ~1000 cases and compared them to ~6400 controls also subjected to WGS. The results identify several previously unknown potential candidate genes for IPAH. The study design is not novel, and neither is the analysis, and the progress from these data will be incremental nevertheless this is still the most comprehensive study of this nature in PAH field and thus an important study in that it advances our understanding of IPAH and provides clues to several new potential candidate genes that can now be further explored.

Comments

The functional data they present overall is appropriate given the focus of the manuscript and what is known about these genes especially GDF2. However, they should avoid using strong language such as “genetic findings strongly suggest that a deficiency in GDF2/BMPR2/ACVRL1 signaling in pulmonary artery endothelial cells is critical in PAH pathobiology” As stated above they have identified candidate genes, and they will remain candidate genes till more convincing data are provided. So, use of word “Strong” seems somewhat premature. This needs to be fixed at multiple places in the manuscript.

Given that BMPR2 can bind to multiple BMPs do they expect that a deficiency of BMP9 in humans will cause PAH or will that deficiency be partly compensated by other BMPs?

They should provide a table to show how many subjects have variants in more than one PAH related genes including the new candidate genes.

I am somewhat perplexed that they do not provide any data on noncoding variants (intronic and in promoter regions) found in all the known IPAH candidate genes. For example, the role of non-coding variants has long been discussed in the field, but given the size of the genomic location of BMPR2 (~170-180kb), a study to document these has never been attempted. Since the focus of this study was to identify novel variants, and they did WGS they should provide the non-coding variant data on at least the known PAH genes. Could easily do in a table format as a supplementary table. Given the

availability of gnomAD, they should be able to provide a frequency distribution of these variants as well.

For the new candidate genes, did they notice familial transmission? They had ten related individuals their study.

How do they explain that while the overall incidence of IPAH in general population is 1-2 per million the frequency of mutations in their candidate PAH genes in their control population appears to be much higher $\sim 1/1000$? Are they using the correct control population for comparison? Why not use EXac (obviously would only provide information about coding variants) or better gnomAD?

Reviewer #3 (Remarks to the Author):

In this manuscript by Gräf et al, the authors describe a Case-Control study on pulmonary arterial hypertension (PAH) on 1048 cases and 6385 controls and the identification of 4 novel genes ATP13A3, AQP1, SOX17 and GDF2 causing this disease.

Samples were sequenced with Whole Genome Sequencing.

At first the authors detected samples with deleterious mutations in previously known PAH genes, including BMPR2, ACVRL1, ENG, KCNK3, SMAD9 and TBX14, and removed them from the further analysis to increase power of the statistic. For a distinct form of PAH, called pulmonary veno-occlusive disease or pulmonary capillary haemangiomatosis (PVOD/PCH) the authors showed significant association with mutations in EIF2AK4.

The authors performed structural analysis for the novel genes ATP13A3, AQP1, SOX17 and GDF2, functional analysis on the GDF2 variants and expression analysis on ATP13A3, AQP1 and SOX17.

The manuscript is well written and transparent in methods and materials. The programs for Genotype Calling and Chromosome Copy Number Analysis as well as the statistical programs for Case-Control analysis are state of the art.

Minor Concerns:

The Supplementary Figures start with Figure S3, while the Figures S1 and S2 are missing.

In the manuscript (page 6) they refer Figure S1. Please clarify?

Text of Figure 3 (line 1) The word "excluded" is written twice.

Response to reviewers

Reviewers' comments:

Reviewer #1 (Remarks to the Author):

In this study, Graf S et al. investigated whether rare variants of newly identified genes were found in a population of patients with various PAH types, including sporadic PAH, heritable PAH, and PVD (a rare form of PAH). They sought variants by whole genome sequencing in 1038 index patients with PAH and 6385 controls with other rare diseases. The results substantiate previous findings on the frequency of BMPRII mutations in the various subgroups. In addition, 4.7% of the cohort with PAH had rare causal variations in non-BMPRII genes including TBX4, ENG, ACVRL1, SMAD9, KCNK3, and EIF2AK4.

Finally, 6 patients carried protein-truncating variants (PTVs) in ATP13A3. Excluding previously reported genes, an association was evident with rare variants in AQP1 and SOX17. An analysis of rare missense mutations revealed overrepresentation of rare variants in GDF2 (encoding BMP9), and transfecting cultured cells with these variants decreased the release of BMP9.

These data highlight associations linking common and less common gene variants to PAH. The study deserves credit for providing a comprehensive picture of genes associated with PAH in a mixed population of patients with heritable PAH, non-heritable PAH, and PVD. The results, however, induce some frustration, as the number of genes newly identified as associated with PAH is small, their frequency low, and their causal effect unproven.

Response: We thank the reviewer for the helpful critique. To clarify, the PAH cohort only recruited patients with a diagnosis of idiopathic PAH, familial PAH, PVOD, and patients with a history of drug exposure (the vast majority of which were associated with remote exposure to appetite suppressant drugs, and thus are more likely to be idiopathic PAH with coincidental drug exposures). Thus, the cohort was actually quite homogeneous and did not include patients with other forms of Group 1 PAH. In other words, the PAH cohort was specifically designed to maximise the chances of identifying novel rare genetic variation as part of a large-scale case-control analysis. Due to background heterogeneity even within the diagnostic category of idiopathic and heritable PAH, we recognise that our cohort may not be sufficiently powered to detect multiple novel causal genes. However, we are confident that the genes reported here represent robust findings.

The reviewer also refers to common and less common gene variants. We would like to emphasise that our analysis was confined to ultra-rare variants with a MAF of <1 in 10,000. We did not analyse common or low frequency variation. Indeed an analysis of common variation is the subject of ongoing work. The overrepresentation of rare highly deleterious variants in new genes in the PAH cohort is powerful evidence for a causal relationship with the disease. Nevertheless, we recognise that additional validation of these findings will be required (as in all new genetic findings) to be certain of causality and for subsequent clinical genetic counselling. Recognising this we have modified our statements on causality and now provide additional evidence of likely pathogenicity for SOX17 and AQP1 with the addition of familial co-segregation analyses (see Figure 4).

One possible explanation to the low yield of the study is that the controls were patients with rare diseases other than PAH. This point is discussed at the end of the discussion. It probably limited the power of the study. The same methodology with a different control population may well have produced different results. Are the authors planning such a study to assess this hypothesis?

Response: We agree with the reviewer that it would be worth repeating the analysis in a large healthy control cohort when that becomes available. However, there is currently a paucity of control data available for WGS studies, by contrast to SNP-based GWA studies. As none of the controls in our cohort had a diagnosis of PAH, we do not believe that shared rare genetic variation between PAH and non-PAH groups will have significantly limited the power of the analysis. Indeed, a major strength of our study was that the case-control comparison was performed with data generated on the identical

sequencing platform and using the same pipeline for sequence alignment, variant calling and filtering. Despite the size of our cohort, the full contribution of rare genetic variation to heritable PAH will require sequencing and combined analyses of even larger PAH cohorts in the future. We are now planning to extend recruitment to an international PAH cohort over the next few years and, alongside this, we seek to develop a substantial cohort of healthy control individuals for future gene identification studies.

In the patients with PAH, both the nature and the frequency of the pathogenic variants are consistent with earlier reports. However, the large study population probably has considerable overlap with populations included in previous studies. Thus, it is unsurprising that the genes and their frequencies are similar to those in reported previously. The authors should indicate which patients were included in earlier studies.

Response: As we sought to exclude known mutation carriers from our novel gene identification study, our BRIDGE PAH cohort has very little overlap with previously published studies. The vast majority of cases have never been included in a previous genetic study. In total, seventeen patients recruited from Paris were previously reported, including 4 cases with larger deletions, since this is one of the few centres that routinely screens PAH patients for the presence of mutations in *BMPR2*, *ALK1* and *ENG*. These patients were reported in Girerd *et al.* [Eur Respir J., 2016, PMID:26699722]. We have now included a footnote and citation to indicate that patients with these variants have been reported previously.

Some previously reported gene mutations, such as Alk1, were not identified in this study.

Response: We have used the HGNC approved gene nomenclature in the manuscript. Therefore, *ALK1* has been referred to as *ACVRL1*. We identified 9 individuals with *ACVRL1* mutations in this study as shown in Supplementary Table 3, which represents a similar frequency to previous reports.

The term “causative genes” used in the title is an overstatement. Whether the small number of uncommon variants identified in this study have causal effects is unproven.

Response: As mentioned above, we now provide a further validation for *SOX17* and *AQP1* by providing evidence for co-segregation of pathogenic mutations in families (Figure 4). However, we acknowledge that further validation will come from functional studies and confirmation in independent cohorts and have therefore altered the title, as suggested.

An issue worthy of investigation is whether the newly identified genes are associated with the previously known genes. For example, are some of these novel genes associated with the BMPRII mutation and suspected to increase penetrance?

Response: Our case-control analysis for novel genes specifically excluded individuals with mutations in previously reported PAH genes. In line with this reviewer's suggestion we looked again at the excluded individuals and found no examples of co-occurrence of mutations in new and previously reported genes in our cohort. As shown in Table 2a, we did find one PAH case with deleterious rare variants in both *BMPR2* and *SMAD9*. Thus, the possibility of oligogenic inheritance or genetic modifier effects in PAH remains of interest, but will require a larger cohort to assess the extent to which this occurs.

A few experiments have been done to localize AQP1 and SOX17, and a few studies have assessed the function of ATP13A3. However, the function of these genes in PAH remains unclear.

Response: The results presented in this manuscript are dominated by a large scale discovery effort designed to provide major new insights into the missing heritability in PAH. We provide initial but important functional studies to demonstrate that the new genes are indeed expressed and/or localised to pulmonary vascular cells (*ATP13A3*, *AQP1*, and *SOX17*). In addition we show that *SOX17* expression is significantly higher in PAECs than PSMCs (Figure 8). Since *ATP13A3*, unlike *AQP1* and *SOX17*, has never been studied in vascular cells we provided further functional data to show the impact of loss of *ATP13A3* function in endothelial cells. More definitive studies of mechanism are now in progress but will inevitably take 2-3 years to read out and are therefore not within the scope of this

study. These include the use of patient-derived pluripotent stem cells, CRISPR-Cas9 mediated mutagenesis and knockin mice, as well as specific biochemical assays to address the functional impact of mutations.

Reviewer #2 (Remarks to the Author):

In this report Graf et al. present data from a genomic study of a large cohort of patients with IPAH. They used WGS to sequence ~1000 cases and compared them to ~6400 controls also subjected to WGS. The results identify several previously unknown potential candidate genes for IPAH. The study design is not novel, and neither is the analysis, and the progress from these data will be incremental nevertheless this is still the most comprehensive study of this nature in PAH field and thus an important study in that it advances our understanding of IPAH and provides clues to several new potential candidate genes that can now be further explored.

Comments

The functional data they present overall is appropriate given the focus of the manuscript and what is known about these genes especially GDF2. However, they should avoid using strong language such as “genetic findings strongly suggest that a deficiency in GDF2/BMPR2/ACVRL1 signaling in pulmonary artery endothelial cells is critical in PAH pathobiology” As stated above they have identified candidate genes, and they will remain candidate genes till more convincing data are provided. So, use of word “Strong” seems somewhat premature. This needs to be fixed at multiple places in the manuscript.

Response: In this revision, we provide a further level of validation for *SOX17* and *AQP1* by providing evidence for co-segregation of pathogenic mutations in these genes in families (Figure 4). However, we recognise that additional validation of these findings will be required (as in all new genetic findings) to be certain of causality and to allow inclusion of the new variants as actionable for clinical genetics reporting. Recognising this, we have edited the language throughout the manuscript in keeping with the reviewer’s suggestion and have modified our statements on causality, including the title of the manuscript.

Given that BMPR2 can bind to multiple BMPs do they expect that a deficiency of BMP9 in humans will cause PAH or will that deficiency be partly compensated by other BMPs?

Response: Although *BMPR2* is involved in signalling to many BMPs, the selectivity for a particular BMP ligand is in general conferred by the co-presence of a high affinity type 1 receptor. The remarkable finding from human genetics is that mutations in *BMPR2* (type 2 receptor) and *ACVRL1* (type 1 receptor) can cause PAH. This specifically implicates the *BMPR2/ACVRL1* receptor complex in pathobiology rather than other BMP type 1 receptors. Indeed mutations in other BMP type 1 receptors cause colonic polyposis (*BMPR1A* [*ALK3*]) and brachydactyly (*BMPR1B* [*ALK6*]), but are not found in PAH patients. Of the large family (>20 ligands) of BMPs, only BMP9 (and BMP10) bind with high affinity to the *BMPR2/ACVRL1* receptor complex. The major circulating BMP is BMP9. Thus, it does appear that other BMPs (with the possible exception of BMP10) can not readily compensate for deficiency of circulating BMP9. In future studies we plan to explore the relationship between circulating BMP9 levels and activity and BMP10.

They should provide a table to show how many subjects have variants in more than one PAH related genes including the new candidate genes.

Response: Our case-control analysis for novel genes specifically excluded individuals with mutations in previously reported PAH genes. In line with the suggestions by this reviewer and reviewer #1, we looked again at the excluded individuals and found no examples of co-occurrence of mutations in new and previously reported genes in our cohort. As shown in Table 2a, we did find one PAH case with deleterious rare variants in both *BMPR2* and *SMAD9*. Thus, the possibility of oligogenic inheritance in PAH remains of interest, but was not prominent in our cohort and will require a larger study to address the extent to which this occurs.

I am somewhat perplexed that they do not provide any data on noncoding variants (intronic and in promoter regions) found in all the known IPAH candidate genes. For example, the role of non-coding variants has long been discussed in the field, but given the size of the genomic location of BMPR2 (~170-180kb), a study to document these has never been attempted. Since the focus of this study was to identify novel variants, and they did WGS they should provide the non-coding variant data on at least the known PAH genes. Could easily do in a table format as a supplementary table. Given the availability of gnomAD, they should be able to provide a frequency distribution of these variants as well.

Response: We have considered the reviewer's request for information on non-coding variants very carefully. We could provide a Table such as the one below, which illustrates the number of non-coding variants in previously reported genes that survive filtering by MAF < 1 in 10,000 in the given control data set and CADD deleteriousness scores.

Consequence type	BMPR2	SMAD1	TBX4	KCNK3	CAVI	ENG	SMAD4	SMAD9	ACVRL1	ATP13A3	EIF2AK4	AQP1	SOX17	GDF2
upstream_gene_variant	4	3	8	2	2	0	1	0	1	0	0	1	3	2
5_prime_UTR_variant	1	2	0	1	1	5	2	1	0	0	0	0	0	0
intron_variant	16	10	6	10	7	3	5	8	3	5	6	3	0	0
3_prime_UTR_variant	0	0	0	0	1	0	1	0	0	0	0	0	0	0
downstream_gene_variant	0	1	2	1	0	1	0	0	2	1	0	0	0	0
Total	21	16	16	14	11	9	9	9	6	6	6	4	3	2

However, we would argue that inclusion of these data would add little to the present manuscript. Additional functional annotation and bioinformatic analyses will be required to assess these variants properly, even for the known genes. This is a body of work that we are now undertaking using functional annotation derived from vascular cells. Narrowing the search space for non-coding variation to the endothelial or smooth muscle cell regulome is likely to yield additional candidate variants in regulatory regions adjacent to and distant from previously reported and novel genes. This is a major undertaking that is now underway in our group and will form the basis of a separate manuscript once complete. We have now included a statement in the last paragraph of the revised Discussion mentioning that these approaches will form the basis of future research.

For the new candidate genes, did they notice familial transmission? They had ten related individuals their study.

Response: Of the 10 individuals with a relative included in the discovery cohort, all of these were found to have pathogenic mutations in *BMPR2*. However, for the new candidate genes we have gone back to recruiting centres to find examples of a family history of PAH for some of the new genes. We now include evidence for familial segregation of a *SOX17* nonsense mutation with PAH, based on independent screening of a *BMPR2* mutation-negative family. We demonstrate that the *SOX17* variant arose *de novo* in the proband and was subsequently transmitted to 2 children who died of childhood onset PAH. This family has now been included in the revised manuscript, strengthening the case for a causal role of *SOX17* variants in PAH (Figure 4a). We also identified familial segregation with rare *AQP1* variants in 3 small families, which are presented in the revised manuscript, again adding support to the causal nature of these pathogenic variants in *AQP1* (Figure 4b-d).

We did not find examples of familial segregation with autosomal dominant heterozygous mutations in *ATP13A3* or *GDF2*. Although it is possible that *de novo* mutation and incomplete penetrance of these mutations (similar to *BMPR2* mutations) occurs, this will require additional collections of family members in the future to assess.

How do they explain that while the overall incidence of IPAH in general population is 1-2 per million the frequency of mutations in their candidate PAH genes in their control population appears to be much higher ~1/1000? Are they using the correct control population for

comparison? Why not use EXac (obviously would only provide information about coding variants) or better gnomAD?

Response: The frequency of mutations referred to in Supplementary Table 6 indicate the overall frequency of rare variants across the full length of each gene. A closer examination of these variants has revealed that the majority of variants in our control cohort are likely benign/VUS due to poor evolutionary conservation, location outside of functional domains, etc. Conversely, the rare variants detected in our patient cohort are predicted to be damaging by multiple pathogenicity predictions, typically due to mutation of critical amino acid residues.

With regard to the choice of control population, we refer the reviewer to Supplementary Tables 2, 4 and 11, where the allele frequencies of the rare variants in the candidate genes are provided, along with the frequencies of these variants in ExAC and UK10K. None of these variants are present in UK10K, indicating that the reported variants in the candidate genes are ultra-rare, in keeping with the known incidence and prevalence of idiopathic and heritable PAH in the general population. The small number of variants that are present in ExAC occur at a frequency of <1/10,000, which is within the cut-off that we have used for variant filtering. Of note, ExAC includes 42 individuals with PAH-SSc, hence we have not excluded these variants to account for potential overlap with this cohort.

Reviewer #3 (Remarks to the Author):

In this manuscript by Gräf et al, the authors describe a Case-Control study on pulmonary arterial hypertension (PAH) on 1048 cases and 6385 controls and the identification of 4 novel genes ATP13A3, AQP1, SOX17 and GDF2 causing this disease. Samples were sequenced with Whole Genome Sequencing. At first the authors detected samples with deleterious mutations in previously known PAH genes, including BMPR2, ACVRL1, ENG, KCNK3, SMAD9 and TBX14, and removed them from the further analysis to increase power of the statistic. For a distinct form of PAH, called pulmonary veno-occlusive disease or pulmonary capillary haemangiomatosis (PVOD/PCH) the authors showed significant association with mutations in EIF2AK4.

The authors performed structural analysis for the novel genes ATP13A3, AQP1, SOX17 and GDF2, functional analysis on the GDF2 variants and expression analysis on ATP13A3, AQP1 and SOX17. The manuscript is well written and transparent in methods and materials. The programs for Genotype Calling and Chromosome Copy Number Analysis as well as the statistical programs for Case-Control analysis are state of the art.

Response: We thank the reviewer for the very supportive comments.

Minor Concerns:

The Supplementary Figures start with Figure S3, while the Figures S1 and S2 are missing. In the manuscript (page 6) they refer Figure S1. Please clarify?

Response: This oversight has now been resolved.

Text of Figure 3 (line 1) The word "excluded" is written twice.

Response: Thank you, this has now been corrected.

REVIEWERS' COMMENTS:

Reviewer #1 (Remarks to the Author):

My comments are satisfactorily addressed in the reply from the authors

Reviewer #2 (Remarks to the Author):

The authors have answered most of my questions appropriately. Regarding the table on page 4 of the rebuttal which lists the non-coding variants. While the presentation of this table in Supplementary data would be useful, the table could still be greatly improved by providing locations of these variants. These regions are vast and just listing the number of variants is not quite useful.

Response to reviewers

REVIEWERS' COMMENTS:

Reviewer #1 (Remarks to the Author):

My comments are satisfactorily addressed in the reply from the authors

Thank you.

Reviewer #2 (Remarks to the Author):

The authors have answered most of my questions appropriately. Regarding the table on page 4 of the rebuttal which lists the non-coding variants. While the presentation of this table in Supplementary data would be useful, the table could still be greatly improved by providing locations of these variants. These regions are vast and just listing the number of variants is not quite useful.

We have now added an additional table to the Supplementary Information detailing the requested information about the rare non-coding variant surrounding previously established and novel disease genes. This table has also been added to the Supplementary Data Excel spreadsheet for easier data mining.